# Physiological and Transcriptomic Analysis of Tree Peony (*Paeonia* section *Moutan* DC.) in Response to Drought Stress

**Daqiu Zhao** [1,2], **Xiayan Zhang** [1], **Ziwen Fang** [1], **Yanqing Wu** [1] **and Jun Tao** [1,2,*]

[1]   Jiangsu Key Laboratory of Crop Genetics and Physiology, College of Horticulture and Plant Protection, Yangzhou University, Yangzhou 225009, China; dqzhao@yzu.edu.cn (D.Z.); mx120170610@yzu.edu.cn (X.Z.); m18762324166@163.com (Z.F.); yqwu19880928@126.com (Y.W.)
[2]   Institute of Flowers and Trees Industry, Yangzhou University-Rugao City, Rugao 226500, China
[*]   Correspondence: taojun@yzu.edu.cn; Tel.: +86-514-8799-7219

**Abstract:** Tree peony (*Paeonia* section *Moutan* DC.) is a famous ornamental plant, and *P. ostii* has been used for seed oil production in China because it is rich in α-linolenic acid. *P. ostii* has some resistance to drought, but lack of water can severely hinder its growth and development in arid areas. In order to clarify drought stress induced physiological and molecular changes of *P. ostia*, its physiological and transcriptomic analyses were performed under drought stress, and we found that *P. ostii* leaves drooped significantly 12 days after treatment and observed a significant increase in all detected physiological indices in response to drought treatment except leaf water content, chlorophyll, and carotenoid content. Meanwhile, the activity of three antioxidant enzymes basically increased under drought treatment. Moreover, drought treatment significantly reduced photosynthetic and chlorophyll fluorescence parameters except non-photochemical quenching (qN), and maintained more intact mesophyll cell structures. Additionally, many differentially expressed genes (DEGs) were found by transcriptome sequencing, which play an important role in *P. ostia* drought tolerance by controlling a variety of biological processes, including the reactive oxygen species (ROS) system, chlorophyll degradation and photosynthetic competency, fatty acid metabolism, proline metabolism, biosynthesis of secondary metabolism, and plant hormone metabolism. These results provide a better understanding of *P. ostii* responses to drought stress.

**Keywords:** tree peony; drought; transcriptome; membrane damage; reactive oxygen species (ROS)

## 1. Introduction

Drought stress is a type of water stress that is due mainly to the lack of an effective water supply for plants in soil or air, which affects normal plant growth. At present, the problem of global water shortage is becoming increasingly serious, and drought has become one of the most serious threats to plant growth in the world [1]. Therefore, the effects of drought stress on plant growth and development are also growing. Drought stress directly affects plant morphological structure, especially in the nutritive growth stage [1]. One study reported that the most sensitive parts to drought stress are the leaves [2]. Reduced leaf expansion is beneficial for plants under drought conditions, as reduced leaf area leads to reduced transpiration [3]. Moreover, reducing shoot growth and correspondingly increasing root growth can alleviate the damage to plants under drought stress. This is beneficial for the roots to extract more water from deep soil [3]. In addition, drought stress can also have a significant impact on plants during reproductive growth. Moderate drought can promote flower bud differentiation, but excessive drought stress will reduce the number of flowers and promote the degradation of flower

organs, thus reducing the seed-setting rate [1]. Concurrently, drought stress can also affect plant seeds, resulting in a decline in seed yield and quality [4].

Tree peony (*Paeonia* section *Moutan* DC.) contains nine species, all originating from China [5]. Tree peony is one of the top 10 famous flowers in China, with a high ornamental value, and enjoys the reputation of "national color" and "king of flowers". Modern studies have proved that its root bark mainly contains chemical components such as paeonol, which has hepato- and nephroprotective functions, promotes blood circulation, lowers blood sugar, and shows antibacterial and anti-inflammatory activity [6,7]. In addition, it also has high oil value [8]. The roots of tree peony are fleshy, making them more resistant to drought, but a lack of water will also affect the plant's physiological activity and ornamental value [9]. In recent years, many studies have reported the responses of tree peony to drought stress, all directed at the physiological aspects, such as damage to cell membranes, inhibition of photosynthesis, and increased reactive oxygen species (ROS) content [10,11]. However, the molecular mechanism has not been clarified.

As a deep sequencing technology, RNA-seq can be used to obtain the transcriptome of plants and find many differentially expressed genes (DEGs). For example, transcriptomic analysis of the European beech showed that stress caused upregulation of most of the genes associated with lipid and homeostatic processes, and downregulation of genes associated with oxidative stress [12]. Moreover, the transcriptional profiles of dehydration-responsive genes in *Arabidopsis*, rice, and soybean were reported to be similar [13]. Thus, a transcriptomic analysis of tree peony under drought stress is of great significance, and the results can provide a basis for defense-related gene information from the global perspective [14,15].

Oil tree peony (*Paeonia ostii* Fengdan') is a variant of Yangshan peony that evolved during the medicinal cultivation process. *P. ostii* has good seed yield, oil content, and quality, which are suitable for its promotion [16]. At present, more than 20 provinces and autonomous regions, including Henan, Shandong, Anhui, and Hunan, are vigorously promoting the cultivation of *P. ostii* [17]. Moreover, *P. ostii* has developed root system reducing soil erosion, especially in areas such as sandy wasteland, which will give full play to economic, social, and ecological benefits. *P. ostii* has some resistance to drought stress, but a lack of water will also hinder its growth and development, which has become one of the biggest limitations of its cultivation in arid areas. In order to clarify drought stress induced physiological and molecular changes of *P. ostii*, its physiological and transcriptomic analyses were performed under drought stress, and the comparative analysis allows us to understand its complex molecular mechanisms.

## 2. Materials and Methods

### 2.1. Plant Materials and Treatments

In this study, 3-year-old potted *P. ostii* were used as materials, and the experiments were performed between May and June 2017. The *P. ostii* plants were divided into two groups, each group contained 36 plants. One group was watered normally at 17:00 daily as the control, and another group was treated with natural drought. Leaves were taken separately at 0, 4, 8, and 12 days after treatment. First, photosynthetic characteristics and chlorophyll fluorescence parameters were measured, and then 5 leaves were taken as samples on each plant, and 9 plants were taken at the same time. Finally, the samples were stored at $-80\ °C$.

### 2.2. Measurement of Physiological Indices and Antioxidant Enzyme Activity

Accumulation of hydrogen peroxide ($H_2O_2$) was observed by diaminobenzidine (DAB) staining [18]. Accumulation of superoxide anion free radical ($O_2^-$) was observed with a fluorescent microscope (Axio Imager D2, ZEISS, Wetzlar, Germany) using a reagent kit (Shanghai Haling Biotechnology Co., Ltd., Shanghai, China).

Leaf water content was measured using an oven (Shanghai Jinghong Laboratory Instrument Co., Ltd., Shanghai, China) and balance (Suzhou Science Instrument Co., Ltd., Suzhou, Jiangsu, China). First, the appropriate leaves were weighed and recorded as fresh weight (FW) by using the balance; then, the leaves were put into the oven, 105 °C for 5 min and then 65 °C for more than 2 h; Next, weighed the leaves and recorded the weight as dry weight (DW). Leaf water content = (FW − DW)/FW. Relative electrical conductivity (REC) was measured using the reported method [19]. Malondialdehyde (MDA) content was determined according to the guidelines of a reagent kit from Nanjing Jiancheng Bioengineering Institute, China. Additionally, the contents of chlorophyll, carotenoid, and free proline (Pro) were evaluated according to the reported methods [20].

The activity of 3 antioxidant enzymes including ascorbate peroxidase (APX), superoxide dismutase (SOD) as well as peroxidase (POD) was determined using reagent kits (Suzhou Comin Biotechnology Co., Ltd., Suzhou, China).

### 2.3. Measurement of Photosynthetic Characteristics and Chlorophyll Fluorescence Parameters

Portable photosynthesis system (LI-6400, Li-Cor, Lincoln, NE, USA) was used to determine photosynthetic characteristics at 8:30 am local time. The standard leaf chamber was 2 cm × 3 cm, photosynthetic photon quanta flux density (PPFD) was set at 1000 $\mu mol \cdot m^{-2} \cdot s^{-1}$ using a self-taking red and blue light-emitting diode (LED) source. Net photosynthesis rate ($P_n$), transpiration rate ($T_r$), intercellular $CO_2$ concentration ($C_i$), and stomatal conduction ($G_s$) were also recorded in the system. Subsequently, the chlorophyll fluorescence parameters were measured using a chlorophyll fluorescence spectrometer (Heinz Walz GmbH, Effeltrich, Germany) after plants standing for 2 h in the dark. This system recorded maximum fluorescence (Fm) and actual photosynthetic efficiency of photosystem II (Y(II)), maximum quantum yield of PSII (Fv/Fm), non-photochemical chlorophyll fluorescence quenching (qN = (Fv − Fv')/Fv) and non-photochemical quenching (NPQ = (Fm − Fm')/Fm') were calculated [21,22]. All the parameters were measured on the top leaves of 9 different plants of one group on the same day.

### 2.4. Observation of Anatomy

A transmission electron microscope (Tecnai 12, Philips, Holland) was used to observe the anatomical details of leaves, and specific details are referred to in the reported method [23].

### 2.5. RNA-seq and Sata Analysis

Tree peony has no genome, so we performed transcriptome sequencing on it. Leaves of 12-day drought stress treated plants and the control were used to extract total RNA with a MiniBEST Plant RNA Extraction Kit (TaKaRa, Kusatsu, Japan). Six libraries (Control and Drought, three replicates) were prepared and sequenced by Gene Denovo Biotechnology Co. (Guangzhou, China) using an Illumina HiSeq™ 4000 system (Illumina Inc., San Diego, CA, USA). After raw read filtering, transcriptome de novo assembly was performed using short reads assembling program Trinity [24]. And the resulting sequences from Trinity were called unigenes, and various bioinformatics databases were used for their annotation, including the non-redundant protein (NR), non-redundant nucleotide (NT), Interpro and Gene Ontology (GO), cluster of orthologous groups of proteins (COG), Kyoto Encyclopaedia of Genes and Genomes (KEGG).

The unigene expression was calculated and normalized to reads per kilo bases per million reads (RPKM) [25]. The threshold for significantly differentially expressed genes (DEGs) was set at a fold change ≥2.0 and adjusted *p*-value ≤ 0.05. DEG functions were explored through GO and KEGG pathway analysis and the terms which *Q*-value ≤ 0.05 were defined as significant enriched. This was performed to identify significantly enriched metabolic pathways.

*2.6. Quantitative Real-Time Polymerase Chain Reaction (qRT-PCR) Verification*

In order to verify the reliability of the sequenced data, 18 genes related to drought stress response were selected for quantitative real-time polymerase chain reaction (qRT-PCR) validation. *Ubiquitin* gene (JN699053) was used as an internal reference for this experiment. qRT-PCR was used to analyze gene expression levels with a Bio-Rad CFX Connect$^{TM}$ Optics Module (Bio-Rad, Hercules, CA, USA), and their values were calculated according to the $2^{-\triangle\triangle Ct}$ comparative threshold cycle (Ct) method [26]. All primers used were listed in Supplementary Table S1. The specific details referred to the reported method [27].

*2.7. Statistical Analysis*

All experiments in this study were repeated 3 times randomly, and the variance of the results was analyzed with the SAS/STAT statistical analysis package (version 6.12, SAS Institute, Cary, NC, USA).

## 3. Results

*3.1. Physiological Indices*

When *P. ostii* was stressed with continuous drought, the leaves wilted on day 8 and drooped significantly on day 12 (Figure 1). Subsequently, $H_2O_2$ accumulation was observed using DAB staining, and it showed that $H_2O_2$ accumulation dramatically increased in drought-treated leaves, especially on day 12 (Figure 2A). The accumulation of $O_2^-$ was determined using a fluorescence probe, showing that $O_2^-$ accumulated dramatically in drought-treated leaves, especially on day 12, as the fluorescence intensity was significantly higher than on other days after treatment (Figure 2B). Moreover, leaf water content was determined to decrease significantly with the development of drought stress; it was 86.43% lower in drought-treated leaves than in the control on day 12. Additionally, as the indices reflecting membrane lipid peroxidation, REC and MDA content increased significantly in drought-treated leaves, which was always higher than in the control, especially on day 12, and the REC and MDA content in drought-treated leaves was 4.07 and 1.68 times the control, respectively. A similar tendency was observed in Pro, its content was increased significantly in drought-treated leaves compared to the control, especially on day 12. Additionally, chlorophyll content first presented an uptrend and then a downtrend in drought-treated leaves, and in comparison with the control, it was significantly higher on day 4 and day 8, while the opposite tendency was observed on day 12. Carotenoid content decreased significantly with the development of drought stress, and in comparison with the control, it was 27.56% lower in drought-treated leaves on day 12 (Figure 2C).

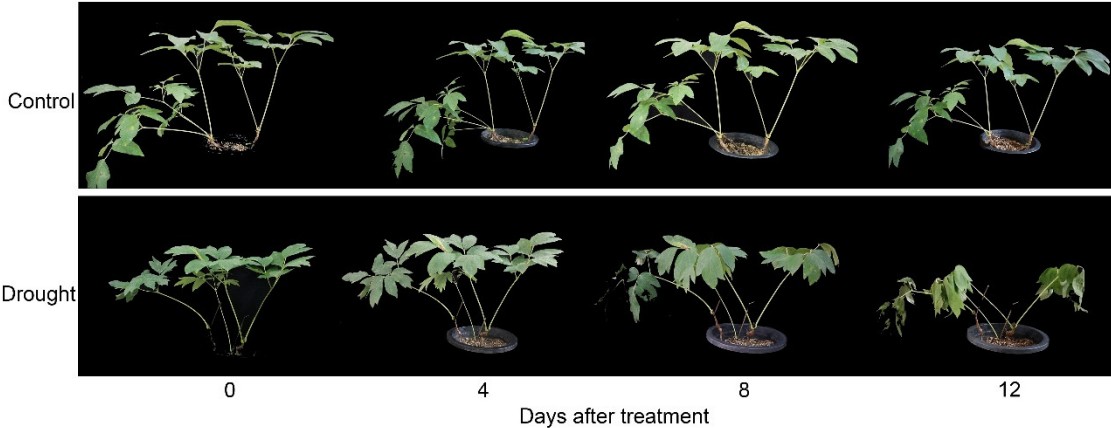

**Figure 1.** Phenotypic changes of drought-treated *P. ostii* and the control.

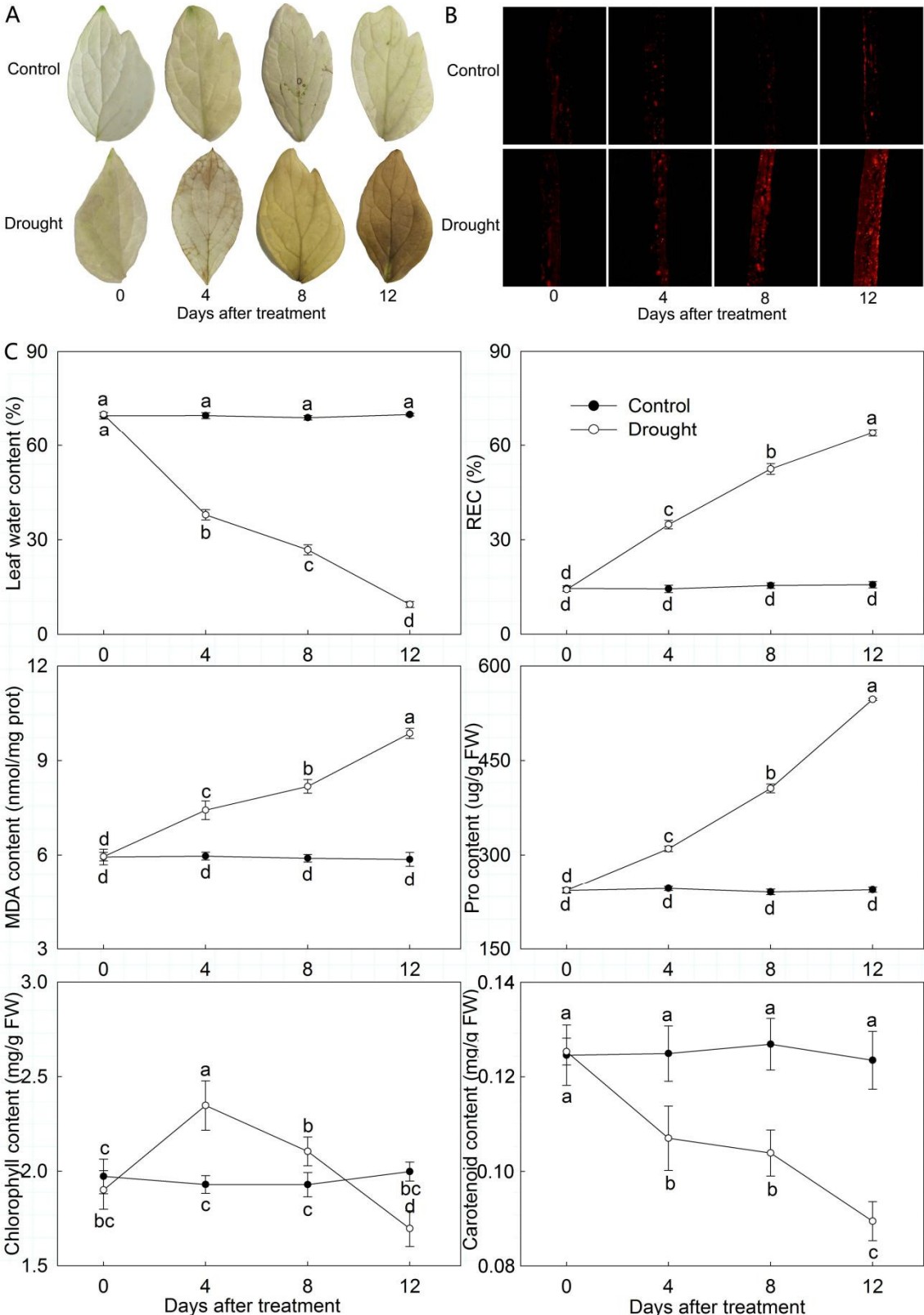

**Figure 2.** Physiological changes of drought-treated *P. ostii* and the control. (**A**) $H_2O_2$ accumulation was detected by diaminobenzidine (DAB) staining. (**B**) $O_2^-$ accumulation was detected by fluorescence probe. (**C**) Other physiological indices. Values represent mean ± standard deviation (SD), and letters indicate significant differences according to Duncan's multiple range test (*p* < 0.05). REC, relative electrical conductivity; MDA, malondialdehyde; Pro, proline; FW, fresh weight.

### 3.2. Antioxidant Enzyme Activity

Among the detected antioxidant enzyme activity, the activity of POD and APX increased significantly when *P. ostii* was stressed with continuous drought. The POD and APX activity in drought-treated leaves was significantly improved compared to the control. Especially on day 12, POD and APX activity in drought-treated leaves was 3.66 and 0.21 times the control, respectively. SOD activity first showed an uptrend and then a downtrend in drought-treated leaves. It was significantly higher than the control on day 4, while the opposite tendency was observed on day 12 (Figure 3).

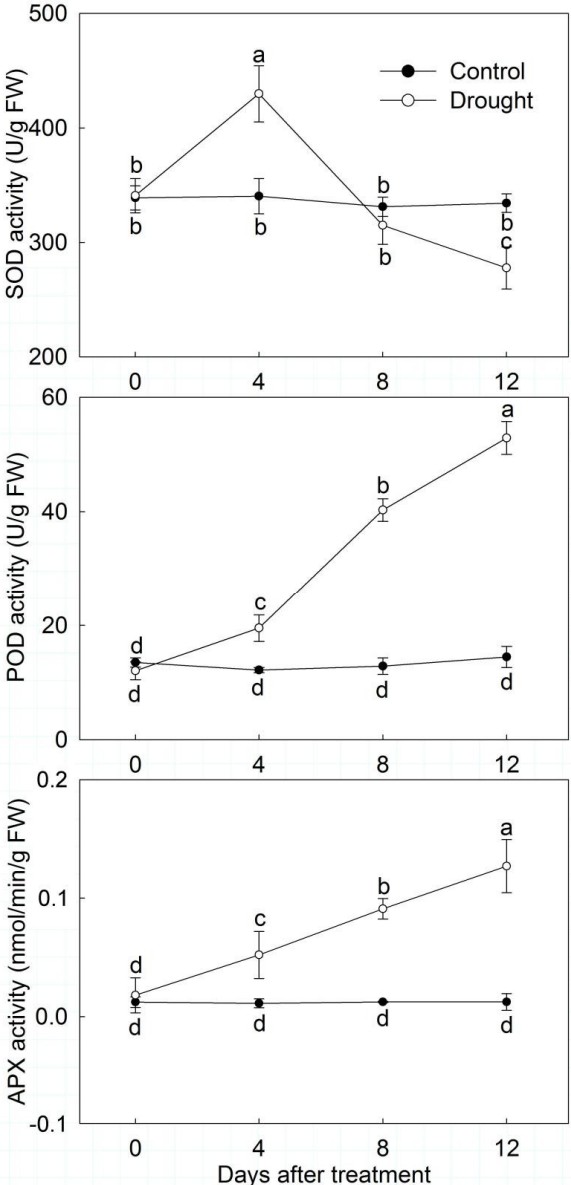

**Figure 3.** Protective enzyme activity changes of drought-treated *P. ostii* and the control. Values represent mean ± standard deviation (SD), and letters indicate significant differences according to Duncan's multiple range test ($p < 0.05$). SOD, superoxide dismutase; POD, peroxidase; APX, ascorbate peroxidase.

### 3.3. Photosynthesis

Drought stress affected the photosynthetic characteristics of *P. ostii* (Figure 4A). $P_n$, $G_s$, $C_i$, and $T_r$ in the control basically remained unchanged, whereas in drought-treated leaves they all showed a downward trend with the development of drought stress. Under drought stress, $P_n$, $G_s$, $C_i$, and $T_r$

significantly decreased by 8.85%, 39.05%, 55.74%, and 17.58%, respectively, on day 12. Moreover, *P. ostii* chlorophyll fluorescence parameters were also significantly affected by drought stress (Figure 4B). Fv/Fm and Y(II) exhibited a downtrend when exposed to drought stress, but qN and NPQ presented the opposite tendency. Fv/Fm and Y(II) in drought-treated leaves were significantly decreased by 62.32% and 10.51% on day 12, respectively. But qN and NPQ was significantly increased by 673.80% and 185.09% on day 12, respectively. Furthermore, Fv/Fm and Y(II) were significantly lower in drought-treated leaves than in the control, and the opposite trend was detected for qN and NPQ.

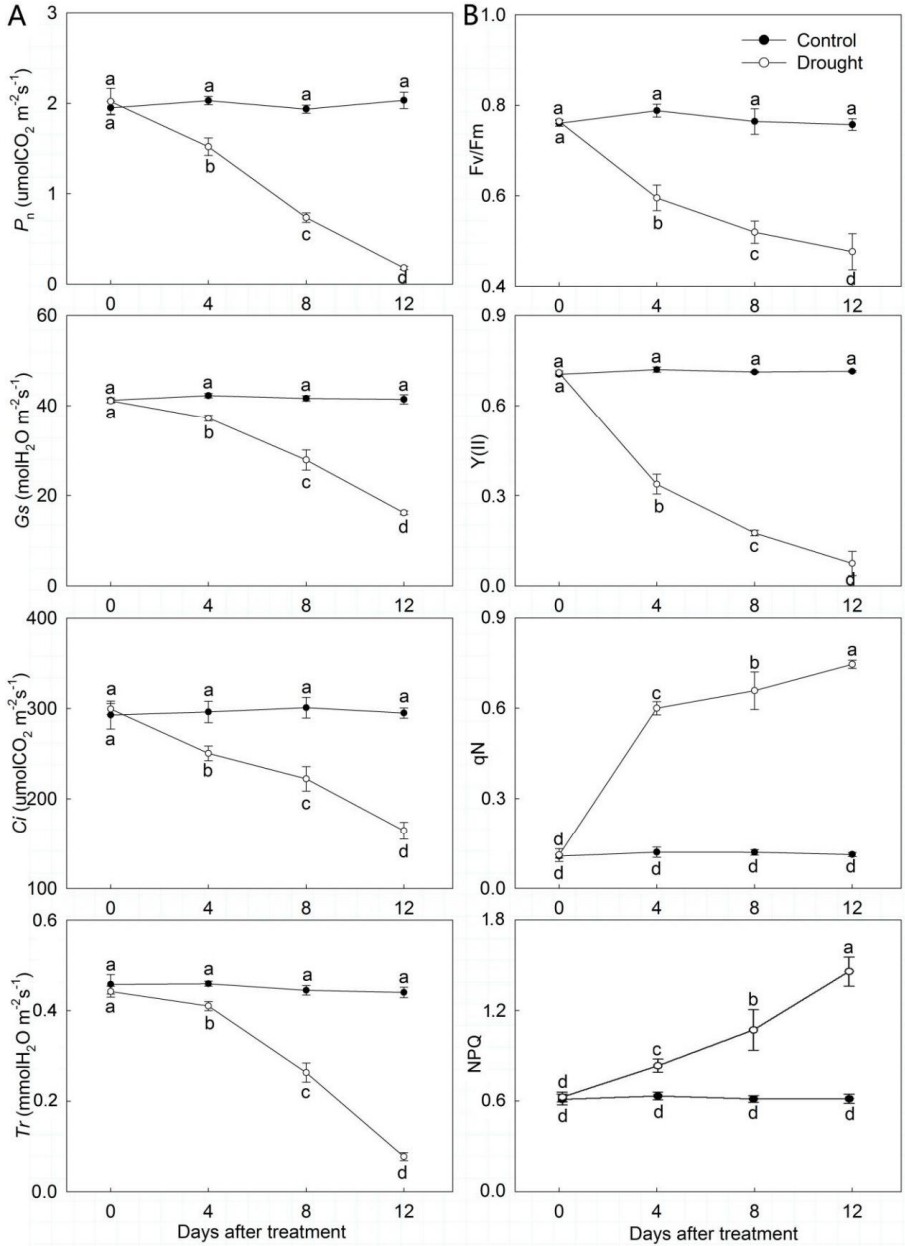

**Figure 4.** Photosynthetic characteristics and chlorophyll fluorescence parameters changes of drought-treated *P. ostii* and the control. (**A**) Photosynthetic characteristics. (**B**) Chlorophyll fluorescence parameters. Values represent mean ± standard deviation (SD), and letters indicate significant differences according to Duncan's multiple range test ($p < 0.05$). $P_n$, Net photosynthesis rate; $T_r$, transpiration rate; $C_i$, intercellular $CO_2$ concentration; $G_s$, stomatal conduction; NPQ, non-photochemical quenching; qN, non-photochemical chlorophyll fluorescence quenching; Fv, variable fluorescence; Fm, maximum fluorescence; Y(II), photosynthetic efficiency of photosystem II.

### 3.4. Anatomy Observation

Drought stress resulted in *P. ostii* mesophyll cell changes. The mesophyll cell ultrastructures of the control and drought-treated leaves on day 0 were very similar. Chloroplasts were the more prominent cell organelles; they were mostly oval in shape and arranged close to the cell membrane in large numbers. Additionally, some starch grains were also observed in some chloroplasts. On day 12, the mesophyll cell ultrastructure of the control remained basically unchanged. However, the chloroplasts in drought-treated leaves had a more rounded shape than previously observed, emerging with large starch grains, and their membranes were blurred (Figure 5).

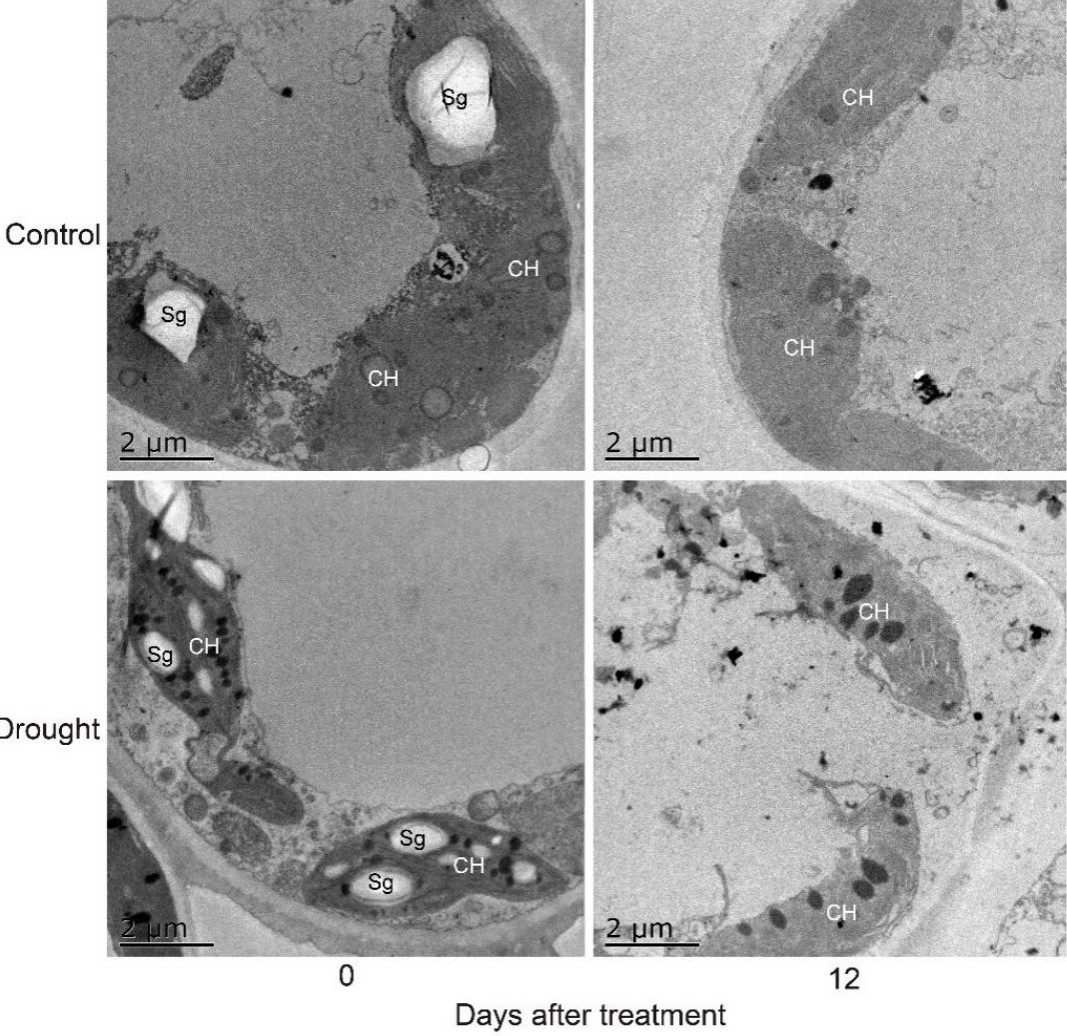

**Figure 5.** Mesophyll cell changes of drought-treated *P. ostii* and the control. CH, chloroplast; Sg, starch grains.

### 3.5. Sequence Analysis, Transcript Assembly, and Gene Functional Annotation

To define drought stress-responsive transcriptome in *P. ostii*, RNA samples from 12-day drought-stressed leaves and the control were used for RNA-Seq. Six libraries were constructed, and an average of 45,588,582 total reads with a single read length of 150 nt, Q30 percentage (the percentage of nucleotides with quality value larger than 20 in reads) of 95.96% and GC percentage (the percentage of G and C bases in reads) of 44.93% were generated from each library (Supplementary Table S2). These data have been deposited in the National Center for Biotechnology Information (NCBI) (SRA: SRP161474). After trimming adapters, filtering out low-quality reads, and de novo assembly,

78,391 unigenes with a mean length of 825 bp and N50 of 1368 across a total of 64,679,096 bp were obtained.

Subsequently, 33,877 unigenes were annotated successfully (Supplementary Figure S1A). A total of 33,594 unigenes displayed significant similarity to the non-redundant protein database (Nr), 9004 unigenes with E values less than 1E-150, 15,322 unigenes with E values between 1E-150 and 1E-20, and 9268 unigene with E values between 1E-20 and 1E-5. Subsequently, to analyze the conservation of sequences, we compared *P. ostii* sequences to those from other species. The top match was grape (*Vitis vinifera* L.) (7174 unigenes), followed by cacao (*Theobroma cacao* L.) (2748 unigenes), lotus (*Nelumbo nucifera* Gaertn.) (1679 unigenes), plum (*Prunus mume* (Siebold) Siebold & Zucc.) (1471 unigenes), barbadosnut (*Jatropha curcas* L.) (1341 unigenes), alfalfa (*Medicago truncatula* Gaertn.) (1175 unigenes), tree cotton (*Gossypium arboreum* L.) (1038 unigenes), sweet orange (*Citrus sinensis* (L.) Osbeck) (945 unigenes), euphrates poplar (*Populus euphratica* Oliv.) (931 unigenes), and rape (*Brassica napus* L.) (882 unigenes). In addition, there were 22,876 unigene annotations to the Swiss-Prot database, 5084 with E values less than 1E-150, 6117 with E values between 1E-50 and 1E-150, and 11,675 with E values between 1E-50 and 1E-5. There were 20,863 unigenes annotated to the cluster of orthologous groups of proteins (COG) database and 10,353 unigenes with E values between 1E-50 and 1E-5. In the KEGG database, 12,726 unigenes were annotated. There were 4325 unigenes (23.99%) with E values less than 1E-150. In addition, 9859 unigenes were annotated with the corresponding functional genes in four databases (Supplementary Figure S1B).

*3.6. Differential Gene Expression Analysis under Drought Stress*

Differential gene expression analysis was conducted between the control and drought group. A total of 22,870 DEGs were expressed under drought stress, with 12,246 of them upregulated and 10,624 downregulated (Figure 6A). Subsequently, 18 gene expression levels were validated by qRT-PCR, and we found a significant positive correlation ($R^2 = 0.948$) between their results and RNA-Seq data (Figure 6B), which revealed that the RNA-seq data were credible. To classify these DEGs functionally, they were annotated to the Gene Ontology (GO) database (Figure 7). They were involved in biological processes (43,734), cellular components (35,652), and molecular function (23,701); those that were closely related to drought stress, such as cellular processes, metabolic processes, cell parts, bindings, and catalytic activities, all had obvious changes in response. Among these, 4143 DEGs were upregulated and 6227 (44%) were annotated to biological processes, of which the more abundant were concentrated in the metabolic and cellular processes. Another 4414 DEGs (32%) were annotated to cellular components, among which the more significant were the cell and cell part, and 3371 (24%) were annotated to the molecular function, the more significant of which were associated with catalytic activity and binding. Of the 17,683 downregulated DEGs, there were 7324 (41%) annotations to biological processes, 6460 (37%) annotations to cellular components, and 3903 (22%) annotations to molecular functions, where the more significant distribution of enrichment was the same as the distribution of upregulated DEGs.

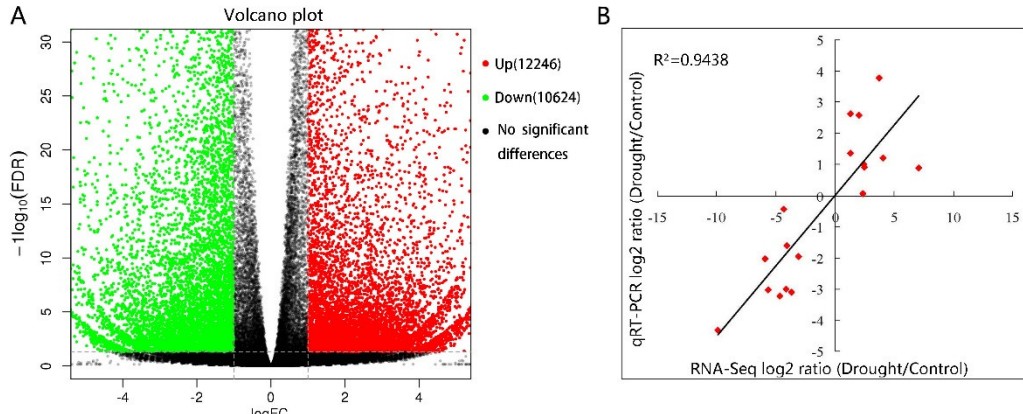

**Figure 6.** Analysis and validation of differentially expressed genes (DEGs) between drought-treated *P. ostii* and the control. (**A**) Volcano plot of DEGs. *x*-axis represents log2 transformed fold change; *y*-axis represents −log10 false discovery rate, red points represent upregulated DEGs, blue points represent downregulated DEGs, and black points represent non-DEGs. (**B**) Correlation of gene expression results obtained from RNA-Seq (*x*-axis) and quantitative real-time polymerase chain reaction (qRT-PCR) (*y*-axis) analysis. Correlation assay performed for 18 DEGs with log2 ratio $\geq 1.00$ or $\leq -1.00$. FDR, false discovery rate; FC, fold change.

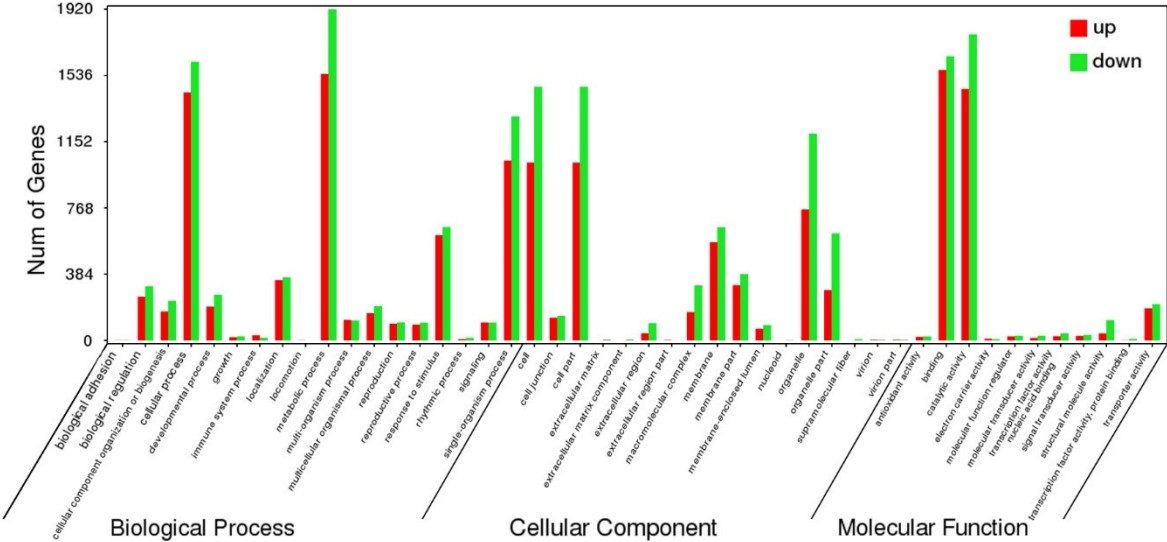

**Figure 7.** Gene Ontology (GO) enrichment classification of differentially expressed genes (DEGs) between drought-treated *P. ostii* and the control. Red histograms represent upregulated DEGs, green histograms represent downregulated DEGs.

KEGG annotation was performed on all DEGs between the control and drought group, and 6997 DEGs were obtained, corresponding to 126 pathways, and only 23 pathways (7 upregulated and 16 downregulated) met a *Q*-value $\leq 0.05$ (Figure 8). To our knowledge, these pathways were divided into six main categories: ROS system (glutathione metabolism); chlorophyll degradation and photosynthetic competency (photosynthesis, photosynthesis-antenna proteins, carbon fixation in photosynthetic organisms, porphyrin and chlorophyll metabolism, glyoxylate and dicarboxylate metabolism, carbon metabolism); proline metabolism (arginine and proline metabolism); biosynthesis of secondary metabolism (flavonoid biosynthesis, stilbenoid, diarylheptanoid and gingerol biosynthesis, phenylpropanoid biosynthesis, carotenoid biosynthesis); fatty acid metabolism (alpha-linolenic acid metabolism, pentose phosphate pathway, fatty acid elongation); and plant hormone metabolism (zeatin biosynthesis). The upregulated DEGs were involved in arginine and

proline metabolism, flavonoid biosynthesis, stilbenoid, diarylheptanoid and gingerol biosynthesis, alpha-linolenic acid metabolism, whereas the down-regulated DEGs were involved in glutathione metabolism, photosynthesis, photosynthesis—antenna proteins, carbon fixation in photosynthetic organisms, porphyrin and chlorophyll metabolism, glyoxylate and dicarboxylate metabolism, carbon metabolism, phenylpropanoid biosynthesis, carotenoid biosynthesis, pentose phosphate pathway, fatty acid elongation, zeatin biosynthesis (Figure 9 and Supplementary Table S3).

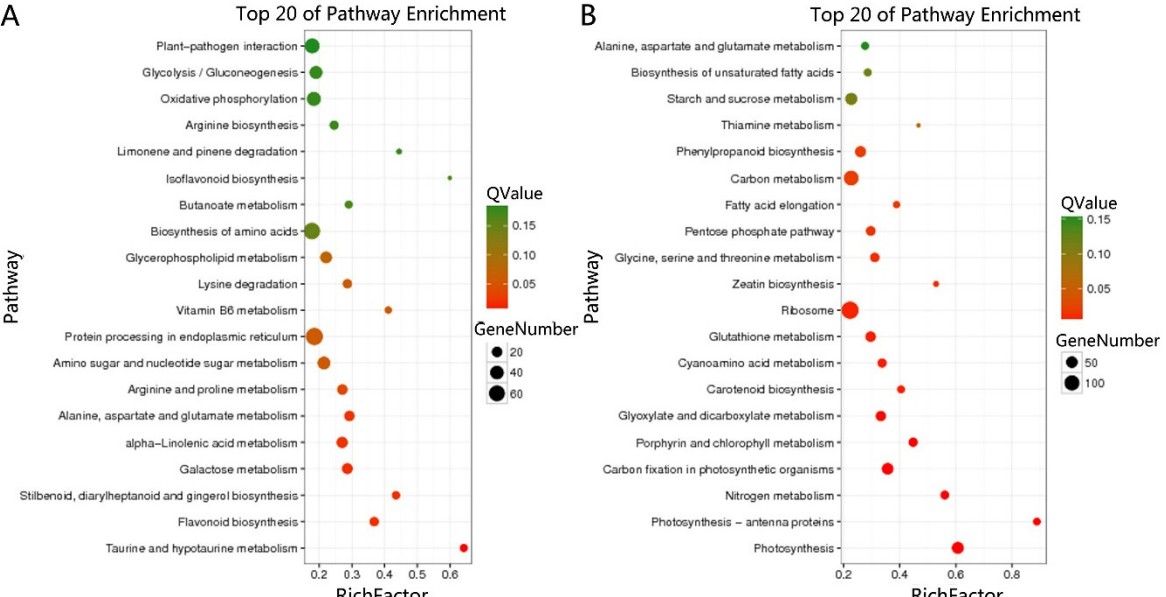

**Figure 8.** Kyoto Encyclopaedia of Genes and Genomes (KEGG) enrichment analysis of differentially expressed genes (DEGs) between drought-treated *P. ostii* and the control. (**A**) Top 20 upregulated pathways. (**B**) Top 20 downregulated pathways.

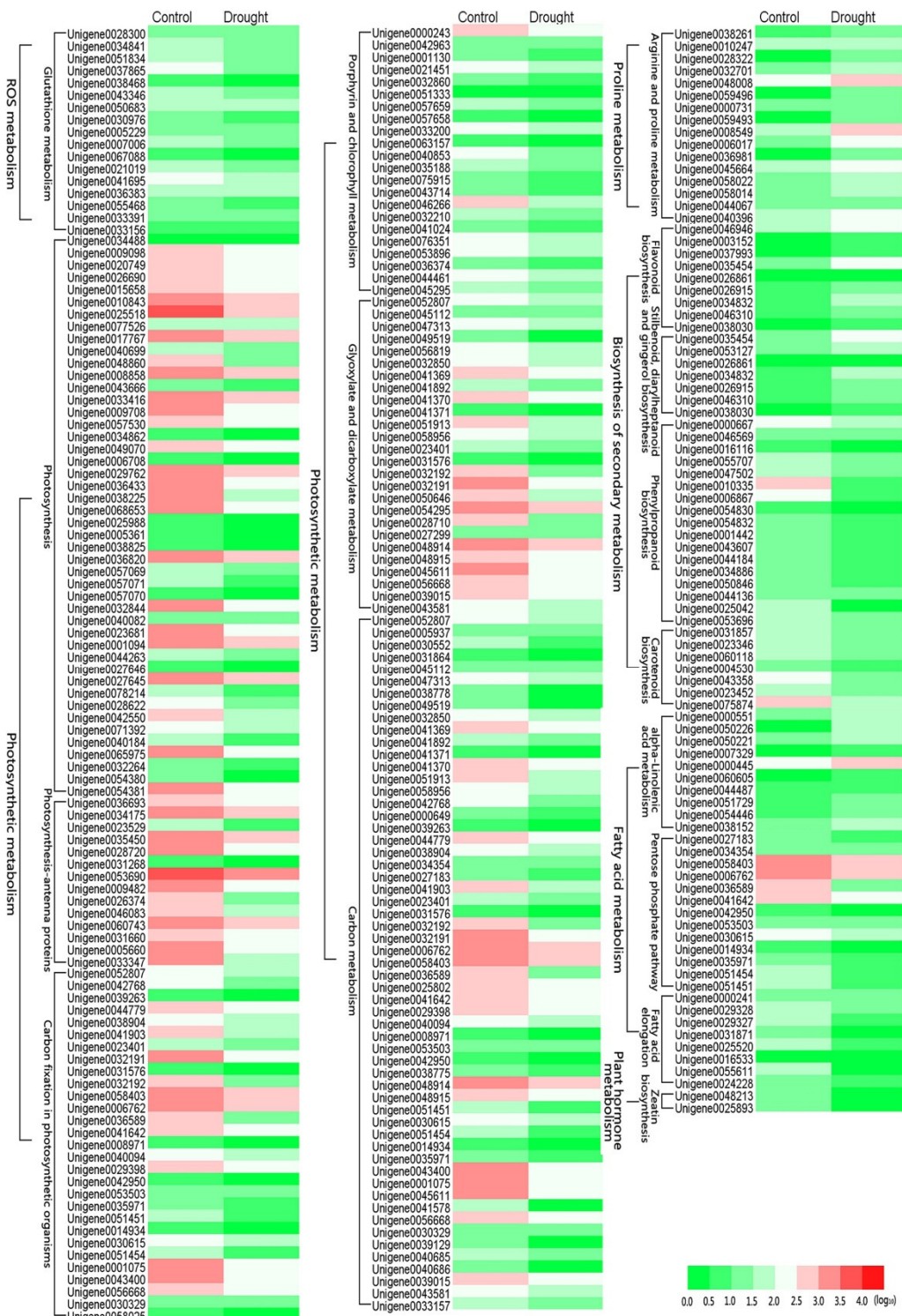

**Figure 9.** Heat map of main differentially expressed gene (DEG) expression patterns involved in reactive oxygen species (ROS) system, photosynthetic metabolism, proline metabolism, biosynthesis of secondary metabolism, fatty acid metabolism and plant hormone metabolism. Annotation information of DEGs can be found in Supplementary Table S3.

## 4. Discussion

As a form of devastating environmental stress in the world, drought affects plant growth and survival. In this study, we carried out drought treatment in *P. ostii*, and it was seen that *P. ostii* leaves

clearly withered on day 8 and drooped significantly on day 12. Moreover, compared with the control, the water content of drought-treated leaves also decreased significantly with the development of drought stress. The leaf is an important organ that consumes water, and its changes under different water supply conditions show the importance of maintaining the water balance in plants [28]. These phenomena are broadly reported, and acutely described in studies on grapevine [29] and wheat [30]. Therefore, *P. ostii* may reduce leaf water dispersion loss by reducing the light-receiving area or the early leaf fall time, so as to resist the drought.

Drought stress can impede some physiological processes of plants, which can damage their normal growth [28]. Maksup et al. [31] reported that drought stress could change plant enzyme activities and ROS accumulation, leading to oxidative damage. ROS include $H_2O_2$ and $O_2^-$ [1]. The contents of $H_2O_2$ and $O_2^-$ in plants can be induced under drought stress [32], which may cause cell membrane damage and lipid peroxidation [1]. Our study observed $H_2O_2$ and $O_2^-$ accumulation of *P. ostii* leaves on different days after drought treatment by DAB staining and a fluorescent probe, and the results showed a significant increase in plant ROS content, especially on day 12, when it was the most different from the control. ROS degrades phospholipids, and damages cell membranes, while REC and MDA can reflect the degree of cell membrane damage [4]. In particular, MDA also reflects the degree of lipid peroxidation [4]. The REC and MDA content of *P. ostii* under drought stress was also gradually increasing in this study. This confirms that the drought stress caused lipid peroxidation of *P. ostii*, destroyed the cell membrane structure, and caused membrane permeability to become large, which confirmed the above viewpoint. This is the same as the results of Cheng et al. [33]. Pro is widely present in plants in a free state and can be used as a cytoplasmic osmotic regulator under stress. In this study, drought stress led to an increase in the Pro content of *P. ostii*, which indicated that *P. ostii* had certain resistance to drought stress.

SOD, POD, and APX are important protective enzyme systems for resisting reactive oxygen species in plant cells, and they have the function of scavenging $H_2O_2$ and $O_2^-$. Gallardo et al. [34] found that the activity of POD and APX scavenging ROS was upregulated under drought stress, which could inhibit membrane lipid peroxidation and membrane structural damage. In this study, POD and APX activity was also observed, and increased significantly under drought stress, the same as the above study. However, SOD activity first showed an uptrend and then a downtrend in drought-treated leaves, consistent with a previous study on potato (*Solanum tuberosum*) [35]. The reason why the SOD activity first rises and then decreases may be because drought stress induces the production of SOD synthesis in the early stage, which leads to the increase of SOD enzyme activity. In the late stage of stress, the severe drought destroys the structure of the synthetase and accelerates the decomposition rate of the enzyme, resulting in decreased SOD enzyme activity.

Photosynthesis is the physiological basis of important life activities and growth of plants. Drought stress can affect changes in plant biomass allocation, which affects photosynthetic parameters. Stomata play an important role in controlling $CO_2$ uptake and water use, and are closely related to photosynthesis and transpiration [36]. In this study, $P_n$ and $T_r$ of drought-treated *P. ostii* were significantly decreased under 12 days of drought stress. The decreased $T_r$ was related to the decrease of $G_s$, which suggested that the drought stress caused the stomatal closure. The stomatal closure also inhibited $CO_2$ absorption, showing decreased $C_i$, which induced the decrease of $P_n$, indicating that drought stress inhibited *P. ostii* photosynthesis by stomatal limitation. Moreover, chloroplasts are the main sites of plant photosynthesis, which also affect $P_n$. Observation of transmission electron microscopy in this study showed that *P. ostii* chloroplast structure was destroyed under drought stress, which revealed that the destroyed chloroplast structure was another reason for the decreased $P_n$. Additionally, drought stress also significantly affected the chlorophyll fluorescence parameters of the plants [37]. The chlorophyll fluorescence-related parameters reflect the photosynthetic reaction and damage of the photosynthetic apparatus. Fv/Fm is an index reflecting the photochemical efficiency of photosynthetic systems. The photochemical efficiency under adverse conditions directly determines the photosynthetic rate of leaves. Meanwhile, qN and NPQ are important indicators of plant self-protection

and have a certain protective effect on photosynthetic organs. An increase in the qN and NPQ values indicates that the protective mechanism such as heat dissipation of the plant has a higher ability to dissipate excess light energy [38]. The drought decreased the Fv/Fm, Y(II) and increased the qN and NPQ of the photosystem II of *P. ostii* leaves, indicating that *P. ostii* protects the reaction center from damage by reducing the capture of light energy and the electron transfer efficiency through photosystem II under drought stress.

Transcriptome has usually been used to study how plants respond to drought stress at the transcriptional level, thereby regulating signal expression and physiological responses to cope with drought stress [39]. In this paper, Illumina HiSeq 4000 high-throughput sequencing technology was used to perform transcriptome sequencing of drought-treated *P. ostii* and the control. More than 43.6 million raw sequencing reads were obtained for each treatment group. After filtering out low-quality reads, the percentage of clean reads obtained in each group accounted for more than 98% of the original reads and the Q30 percentage was also greater than 96%. Wang et al. [30] performed transcriptome analysis on drought-treated loquat with a base Q30 of 92.5%. In comparison to that study, the amount and quality of the data sequenced in this study were relatively higher. Moreover, each *P. ostii* sample had a Q20 ratio greater than 80% and a GC ratio between 35% and 65% [37]. These data results all suggested that the quality of this transcriptome was good.

With regard to the molecular regulation mechanisms of drought stress in *P. ostii*, RNA-Seq in this study yielded 78,391 high-quality unigenes and annotated 33,877 unigenes. Subsequently, 22,870 DEGs were identified, including 12,246 upregulated DEGs and 10,624 downregulated DEGs, and these results were verified by qRT-PCR. In addition, 6997 DEGs were localized to 126 pathways, but there were only 23 pathways (7 upregulated and 16 downregulated) with $Q$-value $\leq 0.05$. They could be divided into six main metabolisms: ROS metabolism, biosynthesis of secondary metabolism, photosynthetic metabolism, proline metabolism, fatty acid metabolism and plant hormone metabolism. According to the differential gene expression in these drought-related pathways, we could understand the transcriptional differences and molecular responses of *P. ostii* under drought stress. During ROS metabolism, DEGs associated with the glutathione metabolic pathway were downregulated, such as glutathione peroxidase gene (*GPX*), glutathione S-transferase gene (*GST*), glutamate-cysteine ligase gene (*GSH*), and spermidine synthase gene (*SPMS*). Glutathione metabolism contributes to the clearance of ROS in *P. ostii* leaves [14], which is inconsistent with the ROS clearance system under drought stress studied by Chaves et al. [40]. This may be due to drought stress exceeding ROS clearance, leading to destruction of glutathione metabolism and increased ROS content. Photosynthesis, photosynthesis-antenna proteins, carbon fixation in photosynthetic organisms, porphyrin and chlorophyll metabolism, glyoxylate and dicarboxylate metabolism, and carbon metabolism pathways were all significantly downregulated in photosynthetic metabolism. Among them, our data showed that drought stress inhibited the chlorophyll biosynthesis process by inhibiting the activity of key enzymes in porphyrin and chlorophyll metabolism, such as magnesium chelatase subunit D (CHLD), magnesium chelatase subunit I (CHLI), magnesium chelatase subunit H (CHLH), magnesium-protoporphyrin IX monomethyl ester (oxidative) cyclase (CRD), and uroporphyrinogen-III synthase (UROS), playing an important role in the synthesis of chlorophyll [41]. Under drought stress, these genes were significantly downregulated, indicating inhibition of the chlorophyll cycle. Moreover, the contents of chlorophyll and carotenoid decreased under drought stress, which was consistent with the transcriptomic data. In addition, photosynthesis-related genes such as PS I, PS II, F-type ATPase, cytochrome b6-f complex, photosynthetic electron transport, and photosynthesis-antenna proteins were significantly downregulated under drought stress, indicating that drought stress had a direct impact on *P. ostii* photosynthesis. In this study, we can see that drought stress affected the photosynthetic characteristics of *P. ostii*. The $P_n$, $G_s$, $C_i$ and $T_r$ in the drought-treated leaves displayed a downward trend with the increase of drought stress, which was consistent with the transcriptome results.

Plants respond to environmental stress by accumulating certain compatible permeates, such as proline, which is known to induce drought tolerance [28]. Moreover, the upregulation of

proline-metabolizing genes and the increase in proline content under drought stress in our experimental results are clear evidence of the induced tolerance of *P. ostii*. Similarly, arginine is converted into orthinine by arginase (upregulated) and then into glutamate-5-semialdehyde (GSA) by the ornithine-δ-aminotransferase (not detected). GSA is then converted into pyrroline 5-carboxylate (P5C) by spontaneous cyclization. Finally, proline is synthesized from the P5C by the P5C reductase (P5CR) enzyme. The biosynthesis of proline begins with the conversion of arginine into orthinine by arginase (ARGAH1) and then into GSA by ornithine-δ-aminotransferase (not detected). GSA is then converted into delta-1-pyrroline-5-carboxylate synthetase (P5CS) by spontaneous cyclization. Finally, proline is synthesized from P5C by P5C reductase (not detected) enzyme [14]. Also, proline contributes to the clearance of ROS [14]. Upregulation of the proline metabolic pathway suggests that proline can contribute to the alleviation of drought stress.

In the biosynthesis of secondary metabolism, many upregulated genes associated with flavonoid biosynthesis and stilbenoid, diarylheptanoid and gingerol biosynthesis were discovered, such as naringenin 3-dioxygenase gene (*F3H*), caffeoyl-CoA *O*-methyltransferase gene (*CCOAOMT*), coumaroylquinate 3'-monooxygenase gene (*CYP98A*) and shikimate O-hydroxycinnamoyltransferase; down-regulated genes associated with phenylpropanol biosynthesis and carotenoid biosynthesis, such as cinnamon-alcohol dehydrogenase gene (*CAD*), beta-glucosidase gene (*GLU*), zeaxanthin epoxidase gene (*ZEP*), 9-cis-epoxycarotenoid dioxygenase gene (*NCED*), 9-cis-beta-carotene 9', 10'-cleaving dioxygenase gene (*CCD*), etc. This indicated that drought stress promoted the biosynthesis of flavonoids and stilbenoid, diarylheptanoid and gingerol, while inhibiting the increase of phenylpropane and carotenoid content.

Fatty acid is a compound composed of carbon, hydrogen and oxygen. It is the main component of neutral fat, phospholipid and glycolipid and its metabolic process greatly affects the content of *P. ostii* seed oil [42]. In fatty acid metabolism, the alpha-linolenic acid metabolism pathway was upregulated and the pentose phosphate pathway and the fatty acid elongation pathway were downregulated. Among them, transketolase gene (*TKL*), ribose 5-phosphate isomerase A gene (*RPI*), fructose-1, 6-bisphosphatase I gene (*FBP*), and other genes in the pentose phosphate pathway can provide NADPH for fatty acid biosynthesis; 3-ketoacyl-CoA synthase gene (*KCS*), enoyl-CoA hydratase gene (*ECH*) and palmitoyl-protein thioesterase gene (*PPT*) in the fatty acid elongation pathway can play a role in the fatty acid biosynthesis pathway [43]. Moreover, drought stress induced an increase in linolenic acid levels, which stimulated the initiation of plant defense mechanisms [44]. In this transcriptome study, the upregulation of genes such as lipoxygenase gene (*LOX*), hydroperoxide dehydratase gene (*CYP*), and 12-oxophytodienoic acid reductase gene (*ORP*) in the alpha-linolenic acid metabolic pathway could also be seen. These all indicate that drought stress leads to a decrease in fatty acid content, and the results are the same as those of Zhang et al. [44].

Additionally, hormones are an important signaling substance in plants that can be induced by environmental factors to regulate leaf wilting [45]. Among the 23 pathways, only zeatin biosynthesis, a pathway associated with plant hormone metabolism, was downregulated. Zeatin is a natural cytokinin that can maintain leaf function and inhibit leaf wilting [46]. In the present study, some DEGs involved in zeatin biosynthesis were found, including cytokinin dehydrogenase gene (*CKX*), cytokinin synthase gene (*IPT*), and cis-zeatin *O*-glucosyltransferase gene (*ZOG*). Li et al. [46] found that the cytokinin content, including zeatin decreased gradually during wilting and senescence of leaves, which was the same as the transcriptome of our study, indicating that drought stress inhibited zeatin biosynthesis and caused the *P. ostii* leaves to wilt and senescence.

According to the above analysis, drought stress caused the downregulation of ROS clearance, photosynthesis and zeatin synthesis in *P. ostii*, which resulted in ROS accumulation, photosynthesis inhibition and leaf wilting. At the same time, drought stress led to the upregulation of proline and linolenic acid synthesis genes, stimulating plants to initiate defense mechanisms and improve drought resistance.

## 5. Conclusions

Overall, this study is the first to report on a comprehensive physiological and transcriptomic analysis of *P. ostia* in response to drought stress. A large number of physiological indices were obtained to reflect the growth statue of *P. ostia* under drought stress, including increased ROS accumulation and membrane lipid peroxidation, damage chloroplast structure and decrease photosynthesis. Moreover, many responsive transcripts and genes that might play an important role in drought stress of *P. ostia* were identified, especially those involved in the ROS system, chlorophyll degradation and photosynthetic competency, proline metabolism, biosynthesis of secondary metabolism, fatty acid metabolism and plant hormone metabolism, which also revealed that drought stress resulted in ROS accumulation, photosynthesis inhibition and leaf wilting, and stimulated plants to initiate defense mechanisms by the upregulation of proline and linolenic acid synthesis. These results could provide a better understanding of the *P. ostii* response to drought stress and lay a foundation for gene expression profile analysis related to the drought tolerance of plants.

**Supplementary Materials:** The following are available online at http://www.mdpi.com/1999-4907/10/2/135/s1, Table S1. Primers used for qRT-PCR, Table S2. Output statistics of RNA-Seq of drought-treated *P. ostii* and the control, Table S3. Six categories pathways and DGEs, Figure S1. Functional annotation of *P. ostia* unigenes from RNA-Seq.

**Author Contributions:** Conceived and designed the experiments: J.T. and D.Z. Performed the experiments: X.Z., Z.F. and Y.W. Analyzed the data: X.Z. and D.Z. Wrote the paper: D.Z. and X.Z.

**Funding:** This work was supported by the young talent support project of the Jiangsu Provincial Association for Science and Technology, the building project of combined and major innovation carrier of Jiangsu Province (BM2016008), the program of key members of Yangzhou University outstanding young teachers, and the priority academic program development from Jiangsu government.

**Acknowledgments:** The authors wish to thank three anonymous reviewers for their input and constructive criticism received.

**Conflicts of Interest:** The authors declare that they have no conflict of interest.

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
