# Peer review of "Physiological and Transcriptomic Analysis of Tree Peony (Paeonia section Moutan DC.) in Response to Drought Stress"

_forests, doi:10.3390/f10020135_

Round 1
Reviewer 1 Report
Dear authors
the study that you presented has a quite simple design. This is not always a disadvantage if, like in this case, it can provide clear and understandable results.
I add therefore only a few requests for clarification and improvement
INTRODUCTION
- I would provide some more details to the readers in order to understand better the scope of your work. e.g. why is it important to study this species more than another? OR Is Paeonia grown as rainfed or irrigated cultivation?
- When you say that “Paeonia ostii is a cultivar of tree peony wild species, Yangshan peony.” You mean that it originated from selections of Yangshan peony populations?
- What do you mean with “improve the ecological environment”? could you better specify?
M&M
- How many plants did you use?
- How many leaves had the plants at the moment of leaves sampling?
- Were photosynthetic parameters and chlorophyll fluorescence measured in a specific moment in the day? Were they measured always on the same leaves?
FIGURES
- Fig 8 blue histograms appear green
DISCUSSION
- The section on the molecular regulation mechanisms provides a huge amount of data, still its discussion looks more like a detailed description of the results than an interpretation of their meaning. For an outside reader it is difficult to grab the added value of this part of the work without an emphasis on why it was done and what has been achieved.
CONCLUSIONS
- I would clarify why your results are relevant? Do they provide clearer insights on how this species copes with water stress? Is there any difference between the mechanisms observed in this species and those observed in other species? There is some novelty
While congratulating with you I thank you for taking into account these feedbacks
Author Response
Response to Reviewer 1 Comments
Point 1: INTRODUCTION
I would provide some more details to the readers in order to understand better the scope of your work. e.g.
- why is it important to study this species more than another? OR Is Paeonia grown as rainfed or irrigated cultivation?
- When you say that “Paeonia ostii is a cultivar of tree peony wild species, Yangshan peony.” You mean that it originated from selections of Yangshan peony populations?
- What do you mean with “improve the ecological environment”? could you better specify?
Response 1: Thanks for your comments very much. We are very sorry that we have not explained the details in some aspects. We have revised the introduction according to your suggestion.
(1) P. ostii has of high medicinal and oil value. As an emerging woody oil crop,Chinais currently promoting P. ostii planting in 20 provinces and autonomous regions. Moreover, P. ostii is mostly planted in sandy soil areas, and its grown was as rainfed cultivation. Therefore, it is necessary to study this species and enhance drought resistance.
(2) In line 64, “Paeonia ostii is a cultivar of tree peony wild species, Yangshan peony.” has been revised to “Paeonia ostii (P. ostii) is a variant of Yangshan peony that evolved during the medicinal cultivation process.”.
(3) In lines 67-69, “In particular, the use of sand wasteland and suitable forestland will give full range to economic, social and ecological benefits and improve the ecological environment.” has been revised to “Moreover, P. ostii has developed root system reducing soil erosion, especially in areas such as sandy wasteland, which will give full play to economic, social, and ecological benefits.”.
Point 2: M&M
- How many plants did you use?
- How many leaves had the plants at the moment of leaves sampling?
- Were photosynthetic parameters and chlorophyll fluorescence measured in a specific moment in the day? Were they measured always on the same leaves?
Response 2: Thanks for your comments very much, and the questions you mentioned are very meaningful. According to your suggestion, we have revised the material and method section.
(1) In lines 79-81, “Some plants were watered normally at 17:00 daily as the control, and the other plants were treated with natural drought.” has been revised to “The P. ostii plants were divided into two groups, each group contained thirty-six plants. One group was watered normally at 17:00 daily as the control, and another group was treated with natural drought.”.
(2) In line 83, “and then 5 leaves were taken as samples on each plant, and 9 plants were taken at the same time.” has been added before “and then the samples were stored at -80 °C.”
(3) In this study, photosynthetic parameters and chlorophyll fluorescence were measured in a specific moment in the day, and they were measured always on the same leaves. In lines 103-114, “Measurement of Photosynthetic Characteristics and Chlorophyll Fluorescence Parameters” has been described as “Portable photosynthesis system (LI-6400, Li-Cor, Lincoln, NE,USA) was used to determinate photosynthetic characteristics at 8:30 am local time. Standard leaf chamber was 2 cm × 3 cm, photosynthetic photon quanta flux density (PPFD) was set at 1000 μmo1·m-2·s-1 using a self-taking red and blue LED source. Net photosynthesis rate (Pn), transpiration rate (Tr), intercellular CO2 concentration (Ci), and stomatal conduction (Gs) were also recorded in the system. Subsequently, the chlorophyll fluorescence parameters were measured using a chlorophyll fluorescence spectrometer (Heinz Walz GmbH 91090 Effeltrich,Germany) after plants standing for 2 hours in the dark. And this system recorded maximum fluorescence (Fm) and actual photosynthetic efficiency of photosystem II (Y(II)), maximum quantum yield of PSII (Fv/Fm), non-photochemical quenching (qN=(Fv-Fv')/Fv) and quantum yield of regulated non-photochemical energy loss (NPQ=(Fm-Fm’)/Fm’) were calculated [21,22]. All the parameters were measured on the top leaves of 9 different plants one group on the same day.”.
Point 3: FIGURES
- Fig 8 blue histograms appear green
Response 3: Thanks for your comments very much, and we have revised the color of Figure 8, please check below:
Figure 8. Gene Ontology (GO) enrichment classification of DEGs between drought-treated P. ostii and the control. Red histograms represent upregulated DEGs, green histograms represent downregulated DEGs.
Point 4: DISCUSSION
- The section on the molecular regulation mechanisms provides a huge amount of data, still its discussion looks more like a detailed description of the results than an interpretation of their meaning. For an outside reader it is difficult to grab the added value of this part of the work without an emphasis on why it was done and what has been achieved.
Response 4: Thanks for your comments very much, and your suggestion is very good. We added the reasons for the molecular regulation mechanism research and the results obtained in the discussion.
(1) “And according to the differential gene expression in these drought-related pathways, we could understand the transcriptional differences and molecular responses of P. ostii under drought stress.” was added in lines 378-380.
(2) “According to the above analysis, drought stress caused the downregulation of ROS clearance, photosynthesis and zeatin synthesis in P. ostii, which resulted in ROS accumulation, photosynthesis inhibition and leaf wilting. At the same time, drought stress led to the upregulation of proline and linolenic acid synthesis genes, stimulating plants to initiate defense mechanisms and improve drought resistance.” was added in lines 447-451.
Point 5: CONCLUSIONS
- I would clarify why your results are relevant? Do they provide clearer insights on how this species copes with water stress? Is there any difference between the mechanisms observed in this species and those observed in other species? There is some novelty
Response 5: Thank you for your comments very much. According to your suggestion, we have revised the conclusions. In lines 453-464, “Overall, this study is the first report about comprehensive physiological and transcriptomic analysis of P. ostia in response to drought stress. Lots of physiological indices were obtained to reflect the growth statue of P. ostia under drought stress. And many responsive transcripts and genes that might play important roles in drought stress of P. ostia were identifed, especially those involved in ROS system, chlorophyll degradation and photosynthetic competencies, proline metabolism, biosynthesis of secondary metabolism, fatty acid metabolism and plant hormone metabolism. These results would provide a better understanding for P. ostii responsed to drought stress and lay the foundation for gene expression profile analysis related to drought tolerance of plants.” has been revised to “Overall, this study is the first to report on a comprehensive physiological and transcriptomic analysis of P. ostia in response to drought stress. A large number of physiological indices were obtained to reflect the growth statue of P. ostia under drought stress, including increase ROS accumulation and membrane lipid peroxidation, damage chloroplast structure and decrease photosynthesis. Moreover, many responsive transcripts and genes that might play an important role in drought stress of P. ostia were identified, especially those involved in the ROS system, chlorophyll degradation and photosynthetic competency, proline metabolism, biosynthesis of secondary metabolism, fatty acid metabolism and plant hormone metabolism, which also revealed that drought stress resulted in ROS accumulation, photosynthesis inhibition and leaf wilting, and stimulated plants to initiate defense mechanisms by the upregulation of proline and linolenic acid synthesis. These results could provide a better understanding of P. ostii respons to drought stress and lay a foundation for gene expression profile analysis related to drought tolerance of plants.”. Thanks very much again.

Reviewer 2 Report
Zhao et al. presents a study of the Peony tree responses to a 12-days drought, involving physiological (anatomical characterization, biophysical measurement of photosynthesis parameters, ROS production and metabolism) and transcriptomic analyses. The broader cultivation of this ornamental plant, currently envisaged for oil production and high-value chemical compounds for human healthcare is well documented in the introduction.This manuscript is interesting and presents a lot of data, but the presentation, including the use of english grammar and analysis of the data and some of the data itself requires a thorough revision.
Major comments:
- In Figure 4b, dissipative mechanisms are estimated via the ‘qN’ parameter (given by the PAM device). It is not clear what measure is being used and if it is reversible or non-reversible NPQ which is being measured here. Much more attention has to be given to the measurements made and not just assigning the abbreviations given by the PAM device. The non-photochemical quenching of the Fo level of fluorescence: Line 326 states « decreased Fo indicated that the PSII reaction center was destroyed or reversibly inactivated « . No, Decrease in Fo suggests a quenching at the level of the PSII core or reduction in chlorophyll content, it is an INCREASE in Fo that suggests irreversible inactivation. Reliable measurements of the non-photochemical quenching can be made with the ‘NPQ’ parameter (NPQ=(Fm-Fm’)/Fm’), independent of Fo evolution. The other photosynthetic measurements relating to Ci etc are also poorly explained. The methods section needs to have more explanation. The results relating to physiology are poorly discussed in the discussion : Lines 330-332: authors conclude from the concomitant decrease in Fv/Fm and increase in qN that the ‘photosynthetic apparatus had not suffered much irreversible damage’. This is misleading, nothing indicates the PSII inhibition is reversible. Destructive non-photochemical quenching is also at play (photoinhibition, qI), and would produce the same effect on Fv/Fm value. I thus recommend considering an additional experiment involving the measurement of the same parameters while rewatering the plants.
- In Figure 5, micrographs are not very convincing. ‘Control-day 12’ does not seem to represent the same cell type or scale than other pictures. Also, lack of contrast on ‘Drought-day 12’.
- the transcriptomic section is incomprehensible but the quality of the data and some of the findings are definitely interesting and is worthwhile to rewrite. We do not know if genome data exist for Peony trees already and if the authors have used a reference genome or transcriptome to map their data. How many gene models are estimated ? What is « the percentage of filtered Q30 « ? It is OK to speak like that in a genomics paper but in the discussion of a manuscript on drought stress for « forest » it is inappropriate. It is not clear : much of what is in the results section for the transcriptomics should be part of the methods section and then a clear analysis of the DEGs can be presented.
- Which DEGs were used for the RT-PCR analysis shown in figure 7B ? There is no list showing the genes that were tested. Why didn’t you test the APX, SOD and POD (btw what is POD ? I couldn’t find anywhere the abbreviation for this enzyme) transcripts as you have nice functional data in figure 3 ?
Minor revision and Other issues:
- NO abbreviations should be used without first explaining what the abbreviation is and given the definition of the parameter.
- Phrasing needs proofreading by a native English writer.
- Too many sentences start with ‘And,’
- Heavy repetition of ‘In comparison with the control’
- Syntax is sometimes clumsy. A few examples: « P. ostii was used as the materials to study its physiological indices » (l. 69), « growth destroyed » (l. 284), « During ROS metabolism, down regulation […] was discovered »
- line 84: « leaf water content was measured using the oven (REF) and balance (REF): technical details (drying time and temperature) would be more useful than the brands of the devices.
- line 89: add reminder of the reason behind measuring proline content (only explained in Discussion section)
- line 169: ‘Compared with the control, Pn, Gs, Ci and were lower, except on day 0’: this is trivial, plants were then in the same conditions.
- line 280: « leaf water content […] decreased significantly with drought stress »/« these phenomena can also be seen in drought-treated grapevine and wheat ». Is this remark necessary, this is quite trivial ? Suggestion for a more appropriate phrasing would be « these phenomena are broadly reported, and acutely described in studies on grapevine and wheat ».
- Lines 284-290: link is made between experimental measurements of ROS accumulation and the severity of membrane damage (REC). It could use a reminder of the mechanisms of phospholipid degradation by ROS (especially when naming MDA experiments).
- Lines 306-308: explanation for the change of trend in SOD activity: unclear phrasing
- Lines 335: « And transcriptome was usually used to study the response to drought stress in plants. »: to be integrated within another sentence
- add spacing and paragraphs to the discussion. Lines from 345 to 419 form only one block of text.
- Too many transcriptomic-data figures are presented (maybe Figures 6 and 7 could go to Supp data). Figure 8 axis-labels font is too small.
Author Response
Response to Reviewer 2 Comments
Point 1: In Figure 4b, dissipative mechanisms are estimated via the ‘qN’ parameter (given by the PAM device). It is not clear what measure is being used and if it is reversible or non-reversible NPQ which is being measured here. Much more attention has to be given to the measurements made and not just assigning the abbreviations given by the PAM device. The non-photochemical quenching of the Fo level of fluorescence: Line 326 states « decreased Fo indicated that the PSII reaction center was destroyed or reversibly inactivated « . No, Decrease in Fo suggests a quenching at the level of the PSII core or reduction in chlorophyll content, it is an INCREASE in Fo that suggests irreversible inactivation. Reliable measurements of the non-photochemical quenching can be made with the ‘NPQ’ parameter (NPQ=(Fm-Fm’)/Fm’), independent of Fo evolution. The other photosynthetic measurements relating to Ci etc are also poorly explained. The methods section needs to have more explanation. The results relating to physiology are poorly discussed in the discussion: Lines 330-332: authors conclude from the concomitant decrease in Fv/Fm and increase in qN that the ‘photosynthetic apparatus had not suffered much irreversible damage’. This is misleading, nothing indicates the PSII inhibition is reversible. Destructive non-photochemical quenching is also at play (photoinhibition, qI), and would produce the same effect on Fv/Fm value. I thus recommend considering an additional experiment involving the measurement of the same parameters while rewatering the plants.
Response 1: Thanks for your comments very much. We are sorry for our carelessness and misunderstanding. We have revised the description of photosynthetic characteristics and chlorophyll fluorescence parameters measurement, redrawn the Figure 4b, and discussed the results relating to physiology in the discussion.(1) In lines 104-115, “Photosynthetic parameters were determined using a portable photosynthesis system (Li-Cor LI-6400,USA) on a cloudless day. And in this system, net photosynthesis rate (Pn), transpiration rate (Tr), intercellular CO2 concentration (Ci) and stomatal conduction (Gs) were recorded. Moreover, a chlorophyll fluorescence spectrometer (Heinz Walz GmbH 91090 Effeltrich,Germany) was used to measure the chlorophyll fluorescence parameters. And this system recorded minimum fluorescence (Fo), maximum fluorescence (Fm) and non-photochemical quenching (qN), and actual photosynthetic efficiency of photosystem II (Y(II)) and maximum quantum yield of PSII (Fv/Fm) were calculated [21,22].” has been revised to “Portable photosynthesis system (LI-6400, Li-Cor, Lincoln, NE, USA) was used to determinate photosynthetic characteristics at 8:30 am local time. Standard leaf chamber was 2 cm × 3 cm, photosynthetic photon quanta flux density (PPFD) was set at 1000 μmo1·m-2·s-1 using a self-taking red and blue LED source. Net photosynthesis rate (Pn), transpiration rate (Tr), intercellular CO2 concentration (Ci), and stomatal conduction (Gs) were also recorded in the system. Subsequently, the chlorophyll fluorescence parameters were measured using a chlorophyll fluorescence spectrometer (Heinz Walz GmbH 91090 Effeltrich,Germany) after plants standing for 2 hours in the dark. And this system recorded maximum fluorescence (Fm) and actual photosynthetic efficiency of photosystem II (Y(II)), maximum quantum yield of PSII (Fv/Fm), non-photochemical quenching (qN=(Fv-Fv')/Fv) and quantum yield of regulated non-photochemical energy loss (NPQ=(Fm-Fm’)/Fm’) were calculated [21,22]. All the parameters were measured on the top leaves of 9 different plants one group on the same day.”.
(2) In lines 190-195, “Fv/Fm, Y(II) and Fo all exhibited a downtrend when exposed to drought stress, but qN presented the opposite tendency. Fv/Fm, Y(II) and Fo in drought-treated leaves were significantly decreased by 62.32%, 10.51% and 74.20% on day 12, respectively, whereas qN was significantly increased by 673.80% on day 12. Furthermore, Fv/Fm, Y(II) and Fo were significantly lower in drought-treated leaves than in the control, as well as the converse trend was detected for qN.” has been revised to “Fv/Fm and Y(II) exhibited a downtrend when exposed to drought stress, but qN and NPQ presented the opposite tendency. Fv/Fm and Y(II) in drought-treated leaves were significantly decreased by 62.32% and 10.51% on day 12, respectively. But qN and NPQ was significantly increased by 673.80% and 185.09% on day 12, respectively. Furthermore, Fv/Fm and Y(II) were significantly lower in drought-treated leaves than in the control, and the opposite trend was detected for qN and NPQ.”.
(3) In lines 348-358, “Fo is the fluorescence level when the PSII reaction center is completely open after dark adaptation (that is, the PSII reaction center and the electron acceptor are all oxidized to be in a state in which electron and light energy can be completely accepted). In this study, our results showed that a decrease of Fv/Fm was accompanied by a decrease of Fo in drought-treated P. ostii. According to the change in Fo, the state of the PSII reaction center can be inferred and the decreased Fo indicated that the PSII reaction center was destroyed or reversibly inactivated, and drought stress could significantly reduce the photochemical efficiency of P. ostii. Meanwhile, qN is a self-protection mechanism of plants which has a certain protective effect on the photosynthetic apparatus [38]. Its value was gradually increasing in drought-treated P. ostii, which could alleviate the impact of drought stress on photosynthesis as well as the damage of excess light energy to the PSII reaction center, indicating that the photosynthetic apparatus had not been suffered from much irreversible damage.” has been revised to “The chlorophyll fluorescence-related parameters reflect the photosynthetic reaction and damage of the photosynthetic apparatus. Fv/Fm is an index reflecting the photochemical efficiency of photosynthetic systems. The photochemical efficiency under adverse conditions directly determines the photosynthetic rate of leaves. Meanwhile, qN and NPQ are important indicators of plant self-protection and have a certain protective effect on photosynthetic organs. An increase in the qN and NPQ values indicates that the protective mechanism such as heat dissipation of the plant has a higher ability to dissipate excess light energy [38]. The drought decreased the Fv/Fm, Y(II) and increased the qN and NPQ of the photosystem II of P. ostii leaves, indicating that P. ostii protects the reaction center from damage by reducing the capture of light energy and the electron transfer efficiency through photosystem II under drought stress.”.
(4) In Figure 4, Fo had been deleted and NPQ had been added, please see the below:
Figure 4. Photosynthetic characteristics and chlorophyll fluorescence parameters changes of drought-treated P. ostii and the control. (A) Photosynthetic characteristics. (B) Chlorophyll fluorescence parameters. Values represent mean ± SD, and letters indicate significant differences according to Duncan’s multiple range test (P < 0.05).
In addition, we are very sorry that we couldn't make an additional experiment, because it is winter now and plants are dormant without leaves. And we would consider your suggestions. Thanks very much again.
Point 2: In Figure 5, micrographs are not very convincing. ‘Control-day 12’ does not seem to represent the same cell type or scale than other pictures. Also, lack of contrast on ‘Drought-day 12’.
Response 2: Thanks for your comments very much. According to your suggestion, we have revised Figure 5, and the scale was enlarged to facilitate the observation of the chloroplast structure change from day 0 to day 12. And in this figure, the lower right corner is the the chloroplast structure of Drought-day 12. Please check it, thanks very much again.
Figure 5. Mesophyll cell changes of drought-treated P. ostii and the control. CH, chloroplast; Sg, starch grains.
Point 3: the transcriptomic section is incomprehensible but the quality of the data and some of the findings are definitely interesting and is worthwhile to rewrite. We do not know if genome data exist for Peony trees already and if the authors have used a reference genome or transcriptome to map their data. How many gene models are estimated? What is « the percentage of filtered Q30 « ? It is OK to speak like that in a genomics paper but in the discussion of a manuscript on drought stress for « forest » it is inappropriate. It is not clear: much of what is in the results section for the transcriptomics should be part of the methods section and then a clear analysis of the DEGs can be presented.
Response 3: Thanks for your comments very much. According to your suggestion, we have made the following revisions:
(1) There has been no genome data exist for tree peony until now. And in line 120, “Tree peony has no genome, so we performed transcriptome sequencing on it.” had been inserted.
(2) Q30 means that the probability of misidentifying a base is 0.1%, or the correct rate is 99.9%. We have inserted the interpretation of Q30 in line 217. Therefore, "the percentage of filtered Q30" means that the percentage of nucleotides with quality value larger than 20 in reads after filtering out low quality reads. In lines 363-365, “and the percentage of clean reads obtained in each group accounted for more than 98% of the original reads, and the percentage of filtered Q30 was also greater than 96%.” has been revised to “After filtering out low quality reads, the percentage of clean reads obtained in each group accounted for more than 98% of the original reads and the Q30 percentage was also greater than 96%.”.
(3) The 2.5 method section has been changed to the following:
2.5. RNA-seq and Sata Analysis
Tree peony has no genome, so we performed transcriptome sequencing on it. Leaves of 12-day drought stress treated plants and the control were used to extract total RNA with a MiniBEST Plant RNA Extraction Kit (TaKaRa,Japan). Six libraries (Control and Drought, three replicates) were prepared and sequenced by Gene Denovo Biotechnology Co. (Guangzhou, China) using an Illumina HiSeq™ 4000 system (Illumina Inc., San Diego, CA, USA). After raw read filtering, transcriptome de novo assembly was performed using short reads assembling program Trinity [24]. And the resulting sequences from Trinity were called unigenes, and various bioinformatics databases were used for their annotation, including the nonredundant protein (NR, ftps://ftp.ncbi.nlm.nih.gov/blast/db), nonredundant nucleotide (NT, ftps://ftp.ncbi.nlm.nih.gov/blast/db), Interpro and gene ontology (GO, http://www.geneontology.org/), cluster of orthologous groups of proteins (COG, http://www.ncbi.nlm.nih.gov/COG/), Kyoto encyclopaedia of genes and genomes (KEGG, https://www.kegg.jp/).
The unigene expression was calculated and normalized to Reads Per kilo bases per Million reads (RPKM) [25]. The threshold for significantly differentially expressed genes (DEGs) was set at a fold change ≥2.0 and adjusted P-value ≤0.05. DEG functions were explored through GO and KEGG pathway analysis and the terms which q-value ≤ 0.05 were defined as significant enriched. This was performed to identify significantly enriched metabolic pathways.
2.6. Quantitative Real-time PCR (qRT-PCR) Verification
In order to verify the reliability of the sequenced data, 18 genes related to drought stress response were selected for qRT-PCR validation. Ubiquitin gene (JN699053) was used as an internal reference for this experiment. qRT-PCR was used to analyze gene expression levels with a Bio-Rad CFX ConnectTM Optics Module (Bio-Rad,USA), and their values were calculated according to the 2-△△Ct comparative threshold cycle (Ct) method [26]. All primers used were listed in Supplementary Table S1. The specific details referred to the reported method [27].
Point 4: Which DEGs were used for the RT-PCR analysis shown in figure 7B? There is no list showing the genes that were tested. Why didn’t you test the APX, SOD and POD (btw what is POD? I couldn’t find anywhere the abbreviation for this enzyme) transcripts as you have nice functional data in figure 3?
Response 4: Thank you very much for your comments. We are sorry that we have not described the name of these DEGs in details, and we have listed gene ID and gene name in Supplementary Table S1, please check below. And these DEGs were selected from 23 pathways that met Q-value ≤0.05 in transcriptome sequencing results, and APX, SOD and POD transcripts had not been found in them. In addition, POD is an abbreviation for peroxidase and can be found in line 101. Thanks very much again.
Supplementary Table S1. Primers used for qRT-PCR.
Gene ID | Gene name | Forward primer (5' - 3') | Reverse primer (5' - 3') |
JN699053 | Ubiquitin | GACCTATACCAAGCCGAAG | CGTTCCAGCACCACAATC |
Unigene0034832 | caffeoyl-CoA O-methyltransferase gene | CGTGAAGTAACCGCAAAA | CAGAGCAGTAGCAAGGAG |
Unigene0028465 | cytokinin dehydrogenase gene | CGGAGGGAGAAGTGTTTT | CTCGTGTAATGGGGTAGG |
Unigene0025893 | adenylate isopentenyltransferase gene | GAAGCATTCTTGAAACACG | TTGACACCACCTTTAGCG |
Unigene0026048 | cis-zeatin O-glucosyltransferase gene | TTAGTAGAAGGTCCATACATCG | GTCAAGCCATTCCAAGCA |
Unigene0023671 | beta-glucosidase gene | ACAGCCAATGAAGGATACC | AACCCATTCTTCCAGTCC |
Unigene0026374 | light-harvesting complex II chlorophyll a/b binding protein 2 gene | AGGCTGGCAATGACTTCT | TGCGTTACAGGGTTACAAA |
Unigene0036589 | fructose-bisphosphate aldolase, class I gene | GAGATGTGCTGCTGCTAC | GCTGTCAGATCCTGGTGT |
Unigene0030552 | D-3-phosphoglycerate dehydrogenase gene | GGGCTTGGTATGAACGTG | ATGTAGCAGAGGTAAGAGGC |
Unigene0014015 | phosphoenolpyruvate carboxykinase (ATP) gene | ACACTTCCTCTTCCTTTGG | GCATTTGGGATGTACTCG |
Unigene0008549 | delta-1-pyrroline-5-carboxylate synthetase gene | TCGGACTAGGTGCTGAGGT | GCAACGGGCATCTGTTAT |
Unigene0036981 | polyamine oxidase gene | GCAGAAGTTGAGCCCATA | GAGACGATTGTCCAGGATT |
Unigene0046310 | coumaroylquinate (coumaroylshikimate) 3'-monooxygenase gene | CAATGCCAGTGTCAAGATAG | AACGGTAGTAGCCGAAAA |
Unigene0038030 | shikimate O-hydroxycinnamoyltransferase gene | AGTTGGCAAGGAGTAGGT | AGAATGGAGCAAGAAAGG |
Unigene0025306 | acyl-CoA oxidase gene | AAACATACACGACCAAGGG | CACAAGCATGATAGTCCAAA |
Unigene0050226 | 12-oxophytodienoic acid reductase gene | CAATGACTTTAGACTTGCTGC | AGGTCCACCGTATTCGTC |
Unigene0000445 | acetyl-CoA acyltransferase 1 gene | GTTGACCCTAAAACTGGAGA | CAGTAGTAGTCCCATCTTTCTT |
Unigene0054446 | alcohol dehydrogenase class-P gene | GCTAGAATTGCAGGTGCT | TGTCATAGTCCTTTGGGTTC |
Unigene0055611 | 3-ketoacyl-CoA synthase gene | GGTCAGGCTCAAGTATGTCA | TGAGGATTTCTTCGGGTC |
Point 5: NO abbreviations should be used without first explaining what the abbreviation is and given the definition of the parameter.
Response 5: Thank you for your comments very much. We have deleted the section of Abbreviations. Thanks for your comments again.
Point 6: Phrasing needs proofreading by a native English writer.
Response 6: Thanks for your comments very much. “MDPI” has been invited to revise the language of this manuscript recently, please see the below:
Point 7: Too many sentences start with ‘And,’.
Response 7: Thanks for your comments very much, and we are very sorry for our poor English. These sentences start with ‘And,’ have been edited by “MDPI”, thanks for your comments again.
Point 8: Heavy repetition of ‘In comparison with the control’.
Response 8: Thanks for your comments very much, and we are very sorry for our poor English. These sentences start with ‘And,’ have been edited by “MDPI”, thanks for your comments again.
Point 9: Syntax is sometimes clumsy. A few examples: « P. ostii was used as the materials to study its physiological indices » (l. 69), « growth destroyed » (l. 284), « During ROS metabolism, down regulation […] was discovered »
Response 9: Thanks for your comments very much, and we have revised them to the best of our ability. In lines 71-72, “P. ostii was used as the material to study its physiological indices and molecular responses under drought stress” has been revised to “the physiological indices and molecular responses of P. ostii under drought stress were studied”. In line 312, “which can cause their normal growth to be destroyed” has been revised to “which can damage their normal growth”. In lines 382-383, “During ROS metabolism, downregulation of genes associated with the glutathione metabolic pathway was discovered” has been revised to “During ROS metabolism, DEGs associated with the glutathione metabolic pathway were downregulated”.
Point 10: line 84: « leaf water content was measured using the oven (REF) and balance (REF): technical details (drying time and temperature) would be more useful than the brands of the devices.
Response 10: Thank you very much for your comments, and we are sorry that the method of determining the leaf water content has not been clearly described. In lines 90-95, “Leaf water content was measured using the oven (Shanghai Jinghong Laboratory Instrument Co., Ltd., China) and balance (Suzhou Science Instrument Co., Ltd., China).” has been revised to “Leaf water content was measured using an oven (Shanghai Jinghong Laboratory Instrument Co., Ltd., China) and balance (Suzhou Science Instrument Co., Ltd., China). First, the appropriate leaves were weighed and recorded as FW by using the balance; then, put the leaves into the oven, 105℃ for 5 min and then 65℃ for more than 2 h; Next, weighed the leaves and recorded the weight as DW. Leaf water content = (FW-DW)/FW.”.
Point 11: line 89: add reminder of the reason behind measuring proline content (only explained in Discussion section)
Response 11: Thanks very much for your comments, and we added “Pro is widely present in plants in a free state and can be used as a cytoplasmic osmotic regulator under stress. In this study, drought stress led to an increase in the Pro content of P. ostii, which indicated that P. ostii had certain resistance to drought stress.” in lines 323-325.
Point 12: line 169: ‘Compared with the control, Pn, Gs, Ci and were lower, except on day 0’: this is trivial, plants were then in the same conditions.
Response 12: Thanks for your comments very much. This sentence has been deleted, thanks again.
Point 13: line 280: « leaf water content […] decreased significantly with drought stress »/« these phenomena can also be seen in drought-treated grapevine and wheat ». Is this remark necessary, this is quite trivial? Suggestion for a more appropriate phrasing would be « these phenomena are broadly reported, and acutely described in studies on grapevine and wheat ».
Response 13: Thanks very much for your comments and suggestion. Your suggestion was very good and suitable. In line 307, “these phenomena can also be seen in drought-treated grapevine and wheat” has been revised to “these phenomena are broadly reported, and acutely described in studies on grapevine and wheat”.
Point 14: Lines 284-290: link is made between experimental measurements of ROS accumulation and the severity of membrane damage (REC). It could use a reminder of the mechanisms of phospholipid degradation by ROS (especially when naming MDA experiments).
Response 14: Thanks for your comments very much. According to your suggestion, we have added link between them. In lines 317-323, “Concurrently, REC can reflect the degree of cell membrane damage, and the higher the value, the more severe is the damage to the cell membrane [4]. The REC of P. ostii under drought stress was also gradually increasing in this study. This indicated that drought stress damaged the cell membrane structure of P. ostii leading to a large membrane permeability, which confirmed the above viewpoint. MDA level also can reflect the degree of lipid peroxidation and membrane damage [4]. In this study, P. ostii MDA content increased with the increase of drought stress, indicating that drought stress could lead to aggravation of lipid peroxidation and cell membrane damage.” has been revised to “ROS degrades phospholipids, and damages cell membranes, while REC and MDA can reflect the degree of cell membrane damage. In particular, MDA also reflects the degree of lipid peroxidation. The REC and MDA content of P. ostii under drought stress was also gradually increasing in this study. This confirms that the drought stress caused lipid peroxidation of P. ostii, destroyed the cell membrane structure, and caused membrane permeability to become large, which confirmed the above viewpoint.”.
Point 15: Lines 306-308: explanation for the change of trend in SOD activity: unclear phrasing.
Response 15: Thanks for your comments very much, and we are sorry that the explanation for the trend of SOD activity is unclear. In lines 332-336, “The reason why SOD activity first rises and then decreases is probably because many antioxidant enzymes are inducing enzymes. Drought stress factors promote the induction of enzyme synthesis, and also destroy the structure of synthetase, accelerate the decomposition rate of enzymes, and reduce enzyme activity.” has been revised to “The reason why the SOD activity first rises and then decreases may be because drought stress induces the production of SOD synthesis in the early stage, which leads to the increase of SOD enzyme activity. In the late stage of stress, the severe drought destroys the structure of the synthetase and accelerates the decomposition rate of the enzyme, resulting in decreased SOD enzyme activity.”. Thanks again.
Point 16: Lines 335: « And transcriptome was usually used to study the response to drought stress in plants. »: to be integrated within another sentence.
Response 16: Thank you very much for your comments and we have merged these two sentence. In lines 360-362, “Under drought stress, plants cells will sense stress signals through signal transduction, regulate signal expression and physiological responses, and respond to different levels of transcription and translation. Transcriptome was usually used to study the response to drought stress in plants.” has been revised to “Transcriptome has usually been used to study how plants respond to drought stress at the transcriptional level, thereby regulating signal expression and physiological responses to cope with drought stress.”. Thanks again.
Point 17: add spacing and paragraphs to the discussion. Lines from 345 to 419 form only one block of text.
Response 17: Thanks for your comments very much. This paragraph has been divided into four paragraphs, please check them. Thanks again.
Point 18: Too many transcriptomic-data figures are presented (maybe Figures 6 and 7 could go to Supp data). Figure 8 axis-labels font is too small.
Response 18: Thanks for your comments very much. We have increased the font size of the Figure 8 axis label, but it is also small because many DEGs are involved in it. We have checked the transcription data of Figures 6 and 7 in the manuscript, Figure 6 has been revised to Supplementary Figure S1, but Figure 7 is very necessary for the integrity of this entire article, so it could not be revised to Supp data. Thanks again.

Reviewer 3 Report
The paper is well organized but a clear aims of the study is missing in the abstract and in the end of the introduction. Indeed, the authors reports a large amount of data but they do not indicate the main objectives of the paper. The introduction can be acceptedin the present stage since furnishes a general background for the readers of this study. Materials and methods can be accepted although some more details could be furnished (mainly on items 2.2 and 2.3). results seem robust and strongly support the discussion. Moreover, the main conclusion is too general and should be detailed.
Author Response
Response to Reviewer 3 Comments
Point 1: The paper is well organized but a clear aims of the study is missing in the abstract and in the end of the introduction. Indeed, the authors reports a large amount of data but they do not indicate the main objectives of the paper. The introduction can be acceptedin the present stage since furnishes a general background for the readers of this study. Materials and methods can be accepted although some more details could be furnished (mainly on items 2.2 and 2.3). Results seem robust and strongly support the discussion. Moreover, the main conclusion is too general and should be detailed.
Response 1: Thanks for your comments very much, and your suggestion is very good. Therefore, according to your suggestion, we have made the following revisions.
(1) The objective of this study has been added in the abstract and in the last part of the introduction. In lines 14-15, “In this study, physiological and transcriptomic analysis of P. ostia were performed under drought stress,” has been revised to “In order to clarify drought stress induced physiological and molecular changes of P. ostia, its physiological and transcriptomic analyses were performed under drought stress,”. In lines 71-73, “In this study, P. ostii was used as the materials to study its physiological indices and molecular responses under drought stress, and the comparative analysis allows us to understand its complex molecular mechanisms.” has been revised to “In order to clarify drought stress induced physiological and molecular changes of P. ostia, its physiological and transcriptomic analyses were performed under drought stress, and the comparative analysis allows us to understand its complex molecular mechanisms.”
(2) We have provided more details in the materials and methods section. In lines 77-83 (item 2.1), “In this study, 3-year-old potted P. ostii was used as materials, and the experiments were performed between May and June, 2017. Some plants were watered normally at 17:00 daily as the control, and the other plants were treated with natural drought. And the leaves were taken separately on 0, 4, 8 and 12 days after treatment. Firstly, the measurements of photosynthetic characteristics and chlorophyll fluorescence parameters were performed, and then the samples were stored at -80 °C.” has been revised to “In this study, 3-year-old potted P. ostii were used as materials, and the experiments were performed between May and June 2017. The P. ostii plants were divided into two groups, each group contained thirty-six plants. One group was watered normally at 17:00 daily as the control, and another group was treated with natural drought. Leaves were taken separately at 0, 4, 8, and 12 days after treatment. First, photosynthetic characteristics and chlorophyll fluorescence parameters were measured, and then 5 leaves were taken as samples on each plant, and 9 plants were taken at the same time. Finally, the samples were stored at -80 °C.”.
In lines 89-93 (item 2.2), “Leaf water content was measured using the oven (Shanghai Jinghong Laboratory Instrument Co., Ltd.,China) and balance (Suzhou Science Instrument Co., Ltd.,China).” has been revised to “Leaf water content was measured using an oven (Shanghai Jinghong Laboratory Instrument Co., Ltd., China) and balance (Suzhou Science Instrument Co., Ltd., China). First, the appropriate leaves were weighed and recorded as FW by using the balance; then, put the leaves into the oven, 105℃ for 5 min and then 65℃ for more than 2 h; Next, weighed the leaves and recorded the weight as DW. Leaf water content = (FW-DW)/FW.”.
In lines 102-113 (item 2.3), “Photosynthetic parameters were determined using a portable photosynthesis system (Li-Cor LI-6400,USA) on a cloudless day. And in this system, net photosynthesis rate (Pn), transpiration rate (Tr), intercellular CO2 concentration (Ci) and stomatal conduction (Gs) were recorded. Moreover, a chlorophyll fluorescence spectrometer (Heinz Walz GmbH 91090 Effeltrich,Germany) was used to measure the chlorophyll fluorescence parameters. And this system recorded minimum fluorescence (Fo), maximum fluorescence (Fm) and non-photochemical quenching (qN), and actual photosynthetic efficiency of photosystem II (Y(II)) and maximum quantum yield of PSII (Fv/Fm) were calculated [21,22].” has been revised to “Portable photosynthesis system (LI-6400, Li-Cor, Lincoln, NE, USA) was used to determinate photosynthetic characteristics at 8:30 am local time. Standard leaf chamber was 2 cm × 3 cm, photosynthetic photon quanta flux density (PPFD) was set at 1000 μmo1·m-2·s-1 using a self-taking red and blue LED source. Net photosynthesis rate (Pn), transpiration rate (Tr), intercellular CO2 concentration (Ci), and stomatal conduction (Gs) were also recorded in the system. Subsequently, the chlorophyll fluorescence parameters were measured using a chlorophyll fluorescence spectrometer (Heinz Walz GmbH 91090 Effeltrich,Germany) after plants standing for 2 hours in the dark. And this system recorded maximum fluorescence (Fm) and actual photosynthetic efficiency of photosystem II (Y(II)), maximum quantum yield of PSII (Fv/Fm), non-photochemical quenching (qN=(Fv-Fv')/Fv) and quantum yield of regulated non-photochemical energy loss (NPQ=(Fm-Fm’)/Fm’) were calculated [21,22]. All the parameters were measured on the top leaves of 9 different plants one group on the same day.”.
In lines 118-135 (item 2.5), “Leaves of 12-day drought stress treatment and the control were used to extract total RNA by a MiniBEST Plant RNA Extraction Kit (TaKaRa, Japan). Eukaryotic mRNA must undergo a series of processes and was sequenced using Illumina HiSeqTM 4000 by Gene Denovo Biotechnology Co. (Guangzhou,China) [23].
After raw reads filtering, transcriptome de novo assembly was performed using short reads assembling program Trinity [24]. And the resulting sequences of Trinity were called unigenes, and various bioinformatics databases were used for their annotation.
The unigene expression was calculated and normalized to Reads Per kilo bases per Million reads (RPKM) [25]. GO (gene ontology database) functional analysis and KEGG (kyoto encyclopaedia of genes and genomes database) pathway analysis were performed on differentially expressed genes (DEG) based on fold change ≥ 2.0 and adjusted P-value ≤ 0.05.” has been revised to “Tree peony has no genome, so we performed transcriptome sequencing on it. Leaves of 12-day drought stress treated plants and the control were used to extract total RNA with a MiniBEST Plant RNA Extraction Kit (TaKaRa,Japan). Six libraries (Control and Drought, three replicates) were prepared and sequenced by Gene Denovo Biotechnology Co. (Guangzhou, China) using an Illumina HiSeq™ 4000 system (Illumina Inc., San Diego, CA, USA). After raw read filtering, transcriptome de novo assembly was performed using short reads assembling program Trinity [24]. And the resulting sequences from Trinity were called unigenes, and various bioinformatics databases were used for their annotation, including the nonredundant protein (NR, ftps://ftp.ncbi.nlm.nih.gov/blast/db), nonredundant nucleotide (NT, ftps://ftp.ncbi.nlm.nih.gov/blast/db), Interpro and gene ontology (GO, http://www.geneontology.org/), cluster of orthologous groups of proteins (COG, http://www.ncbi.nlm.nih.gov/COG/), Kyoto encyclopaedia of genes and genomes (KEGG, https://www.kegg.jp/).
The unigene expression was calculated and normalized to Reads Per kilo bases per Million reads (RPKM) [25]. The threshold for significantly differentially expressed genes (DEGs) was set at a fold change ≥2.0 and adjusted P-value ≤0.05. DEG functions were explored through GO and KEGG pathway analysis and the terms which Q-value ≤0.05 were defined as significant enriched. This was performed to identify significantly enriched metabolic pathways.”.
In lines 137-142 (item 2.6), “Quantitative real-time PCR (qRT-PCR) was used to analyze gene expression levels with a BIO-RAD CFX ConnectTM Optics Module (Bio-Rad, USA), and their values were calculated referring to the 2-△△Ct comparative threshold cycle (Ct) method [26]. All used primers were listed in Table S1. And the specific details were referring to the reported method [27].” has been revised to “In order to verify the reliability of the sequenced data, 18 genes related to drought stress response were selected for qRT-PCR validation. Ubiquitin gene (JN699053) was used as an internal reference for this experiment. qRT-PCR was used to analyze gene expression levels with a Bio-Rad CFX ConnectTM Optics Module (Bio-Rad,USA), and their values were calculated according to the 2-△△Ct comparative threshold cycle (Ct) method [26]. All primers used were listed in Supplementary Table S1. The specific details referred to the reported method [27].”.
(3) We have provided more details in the conclusion section. In lines 453-464, “Overall, this study is the first report about comprehensive physiological and transcriptomic analysis of P. ostia in response to drought stress. Lots of physiological indices were obtained to reflect the growth statue of P. ostia under drought stress. And many responsive transcripts and genes that might play important roles in drought stress of P. ostia were identifed, especially those involved in ROS system, chlorophyll degradation and photosynthetic competencies, proline metabolism, biosynthesis of secondary metabolism, fatty acid metabolism and plant hormone metabolism. These results would provide a better understanding for P. ostii responsed to drought stress and lay the foundation for gene expression profile analysis related to drought tolerance of plants.” has been revised to “Overall, this study is the first to report on a comprehensive physiological and transcriptomic analysis of P. ostia in response to drought stress. A large number of physiological indices were obtained to reflect the growth statue of P. ostia under drought stress, including increase ROS accumulation and membrane lipid peroxidation, damage chloroplast structure and decrease photosynthesis. Moreover, many responsive transcripts and genes that might play an important role in drought stress of P. ostia were identified, especially those involved in the ROS system, chlorophyll degradation and photosynthetic competency, proline metabolism, biosynthesis of secondary metabolism, fatty acid metabolism and plant hormone metabolism, which also revealed that drought stress resulted in ROS accumulation, photosynthesis inhibition and leaf wilting, and stimulated plants to initiate defense mechanisms by the upregulation of proline and linolenic acid synthesis. These results could provide a better understanding of P. ostii respons to drought stress and lay a foundation for gene expression profile analysis related to drought tolerance of plants.”. Thanks very much again.

Round 2
Reviewer 2 Report
I thank the authors for their careful rewriting of the manuscript, this is an improved version.